# CHK1 controls zygote pronuclear envelope breakdown by regulating F-actin through interacting with MICAL3

Honghui Zhang [ID][1,2,3,4,5,6,7,8,9,12], Ying Cui[1,2,3,4,5,6,7,8,12], Bohan Yang[1,2,3,4,5,6,7,8], Zhenzhen Hou[1,2,3,4,5,6,7,8], Mengge Zhang[1,2,3,4,5,6,7,8], Wei Su[1,2,3,4,5,6,7,8], Tailai Chen [ID][10], Yuehong Bian[1,2,3,4,5,6,7,8], Mei Li[1,2,3,4,5,6,7,8], Zi-Jiang Chen [ID][1,2,3,4,5,6,7,8,10,11], Han Zhao[1,2,3,4,5,6,7,8], Shigang Zhao [ID][1,2,3,4,5,6,7,8 ✉] & Keliang Wu [ID][1,2,3,4,5,6,7,8 ✉]

## Abstract

CHK1 mutations could cause human zygote arrest at the pronuclei stage, a phenomenon that is not well understood at the molecular level. In this study, we conducted experiments where pre-pronuclei from zygotes with CHK1 mutation were transferred into the cytoplasm of normal enucleated fertilized eggs. This approach rescued the zygote arrest caused by the mutation, resulting in the production of a high-quality blastocyst. This suggests that CHK1 dysfunction primarily disrupts crucial biological processes occurring in the cytoplasm. Further investigation reveals that CHK1 mutants have an impact on the F-actin meshwork, leading to disturbances in pronuclear envelope breakdown. Through co-immunoprecipitation and mass spectrometry analysis of around 6000 mouse zygotes, we identified an interaction between CHK1 and MICAL3, a key regulator of F-actin disassembly. The gain-of-function mutants of CHK1 enhance their interaction with MICAL3 and increase MICAL3 enzymatic activity, resulting in excessive depolymerization of F-actin. These findings shed light on the regulatory mechanism behind pronuclear envelope breakdown during the transition from meiosis to the first mitosis in mammals.

**Keywords** CHK1; Pronuclear Envelope Breakdown; Zygote; F-actin; MICAL3
**Subject Categories** Cell Adhesion, Polarity & Cytoskeleton; Cell Cycle; Development

## Introduction

The life cycle of sexually reproducing organisms commences with the fusion of two haploid germ cells: mature eggs and capacitated sperm. Following this fusion, the parental chromosomes combine and undergo their first replication within distinct male and female pronuclei, each enveloped by the pronuclear envelope. Simultaneously, the female and male pronuclei will migrate closer to each other, followed by pronuclear envelope breakdown (PNEB) (Clift and Schuh, 2013). The intricately regulated process of PNEB is a crucial step for the polymerization of parental chromosomes and the successful initiation of the first mitotic division. However, the precise mechanism underlying PNEB remains largely unknown.

In the process of assisted reproduction, about 10–15% of human embryos are blocked in the early embryonic stage (Betts and Madan, 2008) and ~2% fertilized oocytes could not accomplish the first cell division (Zamora et al, 2011). However, the underlying causes of this phenomenon remain incompletely understood, and there are currently no effective prevention or treatment measures available. In the case of embryos from patients with serine/threonine checkpoint kinase CHK1 mutations in its C-terminal conserved motifs, a significant proportion exhibited arrest at the one-cell stage, often accompanied by a failure of pronuclear envelope breakdown (Chen et al, 2022; Zhang et al, 2021). These heterozygous CHK1 mutations changed their subcellular localization, increased their kinase activities, and inhibited CDC25C–CDK1 pathway, thus resulting in cell cycle arrest (Zhang et al, 2021). However, the patients with CHK1 heterozygous mutations have no other developmental abnormalities except female infertility (Gillespie, 2022; Zhang et al, 2021), suggesting that the regulatory function of CHK1 could diverge between fertilized eggs and somatic cells. Furthermore, there is a pressing need for an in-depth and comprehensive investigation into the intricate molecular mechanisms underlying CHK1's orchestration of pronuclear envelope breakdown

[1]Institute of Women, Children and Reproductive Health, Shandong University, 250012 Jinan, China. [2]State Key Laboratory of Reproductive Medicine and Offspring Health, Shandong University, 250012 Jinan, Shandong, China. [3]National Research Center for Assisted Reproductive Technology and Reproductive Genetics, Shandong University, 250012 Jinan, Shandong, China. [4]Key Laboratory of Reproductive Endocrinology (Shandong University), of Ministry of Education, Shandong University, 250012 Jinan, Shandong, China. [5]Shandong Technology Innovation Center for Reproductive Health, 250012 Jinan, Shandong, China. [6]Shandong Provincial Clinical Research Center for Reproductive Health, 250012 Jinan, Shandong, China. [7]Shandong Key Laboratory of Reproductive Medicine, Shandong Provincial Hospital Affiliated to Shandong First Medical University, 250012 Jinan, Shandong, China. [8]Research Unit of Gametogenesis and Health of ART-Offspring, Chinese Academy of Medical Sciences (No.2021RU001), 250012 Jinan, Shandong, China. [9]The Affiliated Suzhou Hospital of Nanjing Medical University, Suzhou Municipal Hospital, Gusu School, Nanjing Medical University, Nanjing, China. [10]Department of Reproductive Medicine, Ren Ji Hospital, Shanghai Jiao Tong University School of Medicine, Shanghai, China. [11]Shanghai Key Laboratory for Assisted Reproduction and Reproductive Genetics, Shanghai, China. [12]These authors contributed equally: Honghui Zhang, Ying Cui. ✉E-mail: zsg0108@sdu.edu.cn; wukeliang@sdu.edu.cn

in fertilized oocytes. Given the observed enhanced cytoplasmic localization resulting from CHK1 mutations (Zhang et al, 2021), we hypothesized that these mutations could significantly impact certain cytoplasmic biological processes within fertilized eggs. To test this hypothesis, we employed pre-pronuclear transfer technology (Wu et al, 2017) to validate our conjecture.

The exploration of CHK1 substrates has revealed potential phosphorylation targets associated with the cytoskeleton (Blasius et al, 2011). In addition, recent research has unveiled CHK1's role in promoting actin patch formation at the chromatin bridge base during cytokinesis by phosphorylating and activating Src kinase (Dandoulaki et al, 2018). These findings collectively suggest CHK1's significant involvement in the dynamic modulation of the cytoskeleton. Furthermore, it is crucial to underscore that mechanical forces exerted by microtubules on the nuclear lamina and membrane can synergistically contribute to nuclear envelope depolymerization (Beaudouin et al, 2002; Margalit et al, 2005; Muhlhausser and Kutay, 2007; Velez-Aguilera et al, 2022). In addition, transiently nucleated F-actin "shell" formation on the inner surface of the nuclear membrane has been shown to drive nuclear membrane fragmentation (Mori et al, 2014), highlighting the indispensable role of the cytoskeleton in the process of PNEB. However, whether CHK1 exerts an influence on the procedure of PNEB through the regulation of the cytoskeleton remains an unexplored area of investigation.

In this study, we have revealed a novel insight into the role of CHK1 mutants in influencing pronuclear envelope breakdown during fertilized egg development, specifically through a previously unexplored pathway involving the cytoskeleton. These CHK1 mutations may exert an effect on PNEB by modulating the activity of MICAL3, ultimately disrupting the integrity of the F-actin meshwork. This discovery contributes to a deeper understanding of the intricate mechanism underlying the regulation of PNEB in mammalian fertilized eggs.

## Results

### High-quality blastocysts can be obtained after replacement of mutant cytoplasm by pre-pronuclear transfer

The majority of blocked fertilized eggs from patients with CHK1 mutations continued to exhibit distinct pronuclei (Fig. EV1A). These mutations, located within the conserved C-terminal motifs of CHK1, exhibited varying degrees of enhanced cytoplasmic localization (Zhang et al, 2021), suggesting potential effects on biological processes predominantly in the cytoplasm. To explore this further, we employed the emerging human pre-pronuclear transfer (PPNT) technology (Wu et al, 2017) as a means to gain insight into the specific pathogenesis of such mutations and to potentially identify novel treatment approaches. In this regard, we conducted maternal pre-pronuclear transfer (PPN) from a patient with CHK1 mutation (p.R379Q) into enucleated fertilized eggs obtained from healthy donors (Fig. EV1C). This procedure resulted in the generation of four reconstructed fertilized eggs (Em-1, Em-2, Em-3, and Em-4), notably two of which (Em-1 and Em-4) demonstrated the ability to undergo division and develop into morphologically normal blastocysts, especially Em-1 (Figs. 1A and EV1B; Appendix Table S1). In contrast, the reconstructed zygotes

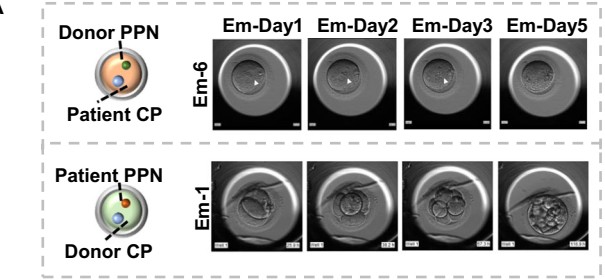

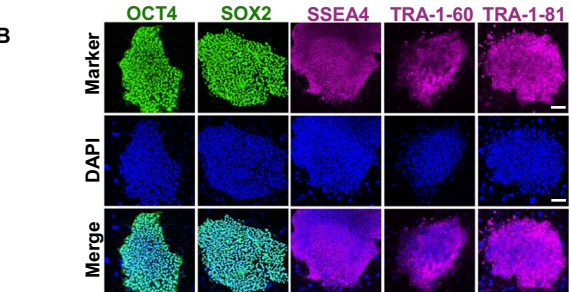

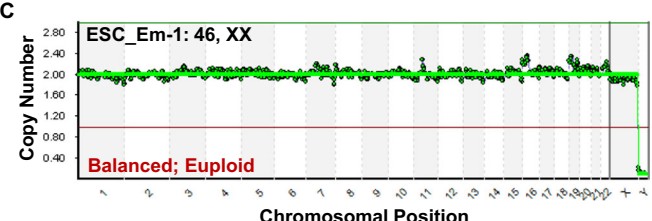

**Figure 1. Successful blastocyst development of CHK1 mutant zygotes through PPNT application.**

(A) Time-lapse recordings of embryo development illustrate the progression of reconstructed embryos. Diagrams on the left depict the reconstruction process: Embryo Em-6 was formed by combining the patient's cytoplasm (CP) with the donor's female pre-pronucleus (PPN), while Embryo Em-1 was constructed using the patient's female pre-pronucleus and the donor's cytoplasm. Pronuclei are indicated by white arrowheads. (B) Pluripotency markers staining of the embryonic stem cell (ESC) line derived from the Em-1 blastocyst. Scale bar: 100 μm. (C) Copy number analysis of the ESC demonstrates its genome integrity. Source data are available online for this figure.

(Em-5, Em-6, and Em-7), comprising mutant cytoplasm and normal PPN, did not exhibit cleavage with distinct pronuclei, even up to the third day of embryo development (Figs. 1A and EV1B; Appendix Table S1).

To assess the developmental potential of the obtained blastocysts, we further established an embryonic stem cell (ESC) line derived from one of the reconstructed blastocysts (Em-1). Immunofluorescence analysis revealed the normal expression of pluripotency markers OCT4, SOX2, SSEA4, TRA1-60, and TRA1-81 in the ESC line (Fig. 1B). In order to further determine the normalcy of the stem cell line, we compared the fluorescence signals with another embryonic stem cell line derived from the patient's blastocyst after treatment with a CHK1 inhibitor (Zhang et al, 2021). As anticipated, the results indicate comparable signals between the two lines (Fig. EV1D,E). Moreover, chromosome copy number (CNV) analysis demonstrated complete genomic ploidy in the established ESC line (Fig. 1C) and the Sanger sequencing results

indicated the ESC line does not harbor the mutated CHK1 allele (Fig. EV1F,G).

In conclusion, the successful acquisition of blastocysts through the replacement of mutant cytoplasm in the patient's zygotes with normal cytoplasm from donated zygotes via PPNT indicates that these CHK1 mutations primarily impact biological processes within the cytoplasm of fertilized eggs. This approach also holds promise as a potential effective treatment strategy for patients with gene mutations primarily influencing cytoplasmic aspects of oocytes or embryos.

## CHK1 mutants disturb the F-actin meshwork in fertilized oocytes

After fertilization, during the mutual migration of male and female pronuclei, a conspicuous aggregation network of F-actin becomes evident in the cytoplasm, particularly surrounding the two pronuclei (Fig. 2A). In line with the observed phenotype in patients with CHK1 mutations (Fig. EV1A), mouse-fertilized eggs over-expressing the mutation (p.F441fs16) arrested in the zygote stage with visible pronuclear membrane clearly indicated by Lamin B1, while wild-type eggs have progressed to the 2-cell stage (Fig. EV2A,B). Considering that p.F441fs*16 works similar to p.R379Q and the former has a strongly destructive effect (Zhang et al, 2021), we therefore mainly select p.F441fs*16 mutation for subsequent experiments. We then introduced mouse-fertilized eggs injected with either wild-type or mutant CHK1 cRNA to assess their F-actin distribution (Fig. 2B). As anticipated, zygotes carrying the CHK1 mutation (p.F441fs16) displayed a significant reduction in F-actin signal, particularly in cytoplasm and around pronuclei, compared to wild-type zygotes (Fig. 2C,D). In addition, consistent with prior findings (Zhang et al, 2021), the overall expression of mutant CHK1 markedly reduced, with an increased proportion exhibiting cytoplasmic localization (Fig. 2C,E). To enhance the clarity of the difference in F-actin quantity and disparities in CHK1 localization, we illustrated the distribution of F-actin signal and EGFP-CHK1 signal through a comprehensive line profile that extends across the pronuclei of zygotes (Fig. EV2C). Collectively, these findings suggest a potential role of the CHK1 mutation in affecting the F-actin meshwork primarily around the pronuclei of fertilized eggs.

## Inhibiting F-actin impairs PNEB of fertilized eggs

The tearing forces of cytoskeleton components on the nuclear lamina and nuclear membrane can synergistically promote depolymerization of the nuclear membrane (Muhlhausser and Kutay, 2007; Velez-Aguilera et al, 2022). Moreover, the transient nucleation of an F-actin "shell" on the inner surface of the nuclear membrane has been shown to trigger its fragmentation (Mori et al, 2014), underscoring the crucial involvement of cytoskeletal elements in nuclear envelope disassembly. Given the observed perturbation in the expression and distribution of F-actin in fertilized eggs carrying CHK1 mutations (Figs. 2C,D and EV2C), we posited that diminished F-actin meshwork might impact the process of PNEB. To investigate this hypothesis, we subjected mouse-fertilized eggs to distinct treatments using microtubule and F-actin inhibitors—Nocodazole and Cytochalasin B—for an 18-hour duration to assess their potential effects on PNEB and subsequent division (Fig. 3A). Remarkably, while the control group

treated with DMSO progressed to the 2-cell stage, neither the Nocodazole nor the Cytochalasin B-treated groups exhibited any signs of cleavage (Fig. 3B,C).

Subsequently, immunofluorescence staining of these arrested zygotes revealed that microtubule inhibition impeded the development of fertilized eggs, yet their pronuclear membranes underwent timely breakdown (Fig. 3D). In sharp contrast, the inhibition of F-actin polymerization using Cytochalasin B led to the retention of distinct pronuclei in blocked fertilized eggs, accompanied by a marked disruption of the F-actin network (Fig. 3D). Consistent with a previous study, another F-actin inhibitor, Cytochalasin D, was shown to delay the onset of PNEB (Scheffler et al, 2021). However, to comprehensively explore the direct potential influence of F-actin inhibition on the PNEB process, distinct from an impact stemming from an earlier step, such as migration, we have strategically administered the F-actin inhibitor primarily starting from phase B, ~2 h prior to the occurrence of PNEB, when the migration phase has past (Fig. EV3A) (Scheffler et al, 2021). As anticipated, the administration of the inhibitor from phase B also effectively obstructed the PNEB process (Fig. EV3B,C). Collectively, these findings provide compelling evidence that the dynamics of F-actin, critically influence the PNEB process in fertilized eggs, suggesting a potential mechanism through which CHK1 mutations may perturb PNEB by mainly affecting F-actin dynamics.

## CHK1 is able to interact with MICAL3

To delve deeper into the mechanisms by which CHK1 regulates the F-actin meshwork in cytoplasm of fertilized eggs, we embarked on an effort to identify a specific target of CHK1 that could act as a crucial intermediary bridge, orchestrating the dynamics of F-actin. To achieve this goal, we meticulously collected 6000 mouse-fertilized eggs for a co-immunoprecipitation (CO-IP) experiment coupled with mass spectrometry. After excluding proteins identified in the IgG control, we successfully pinpointed 32 proteins that exhibit potential interactions with CHK1, among which is MICAL3, a distinct regulator of F-actin (Figs. 4A and EV4G). MICAL3, functioning as a monooxygenase, plays a pivotal role in the oxidation of specific amino acid residues on actin, ultimately leading to the depolymerization of F-actin into actin monomers (Alto and Terman, 2018; Fremont et al, 2017; Kim et al, 2020). Transcriptome data from human preimplantation embryos underscored the prominence of MICAL3's expression in oocytes and embryos, particularly during critical developmental stages such as oocyte, zygote, 2-cell, and 4-cell stages (Fig. EV4A–C) (Yan et al, 2013; Data ref: Yan et al, 2013). Furthermore, MICAL3 predominantly localizes within the cytoplasm of both HEK-293 cells and human oocytes (Fig. EV4D,E), and exhibits co-localization with CHK1 in the human GV oocyte and early embryos (Fig. EV4F). These robust findings collectively underscore the significant role that MICAL3 is likely to play in early preimplantation embryonic development.

To validate the interaction between CHK1 and MICAL3, we initially scrutinized the protein–protein interaction using partial crystal structures of CHK1 (1–307 aa; pdb2qhn) and MICAL3 (1–700 aa; pdb6ici) obtained from the PDB database. Our analysis revealed nonpolar interactions between these components (Appendix Fig. S1). Subsequently, we conducted CO-IP experiments in

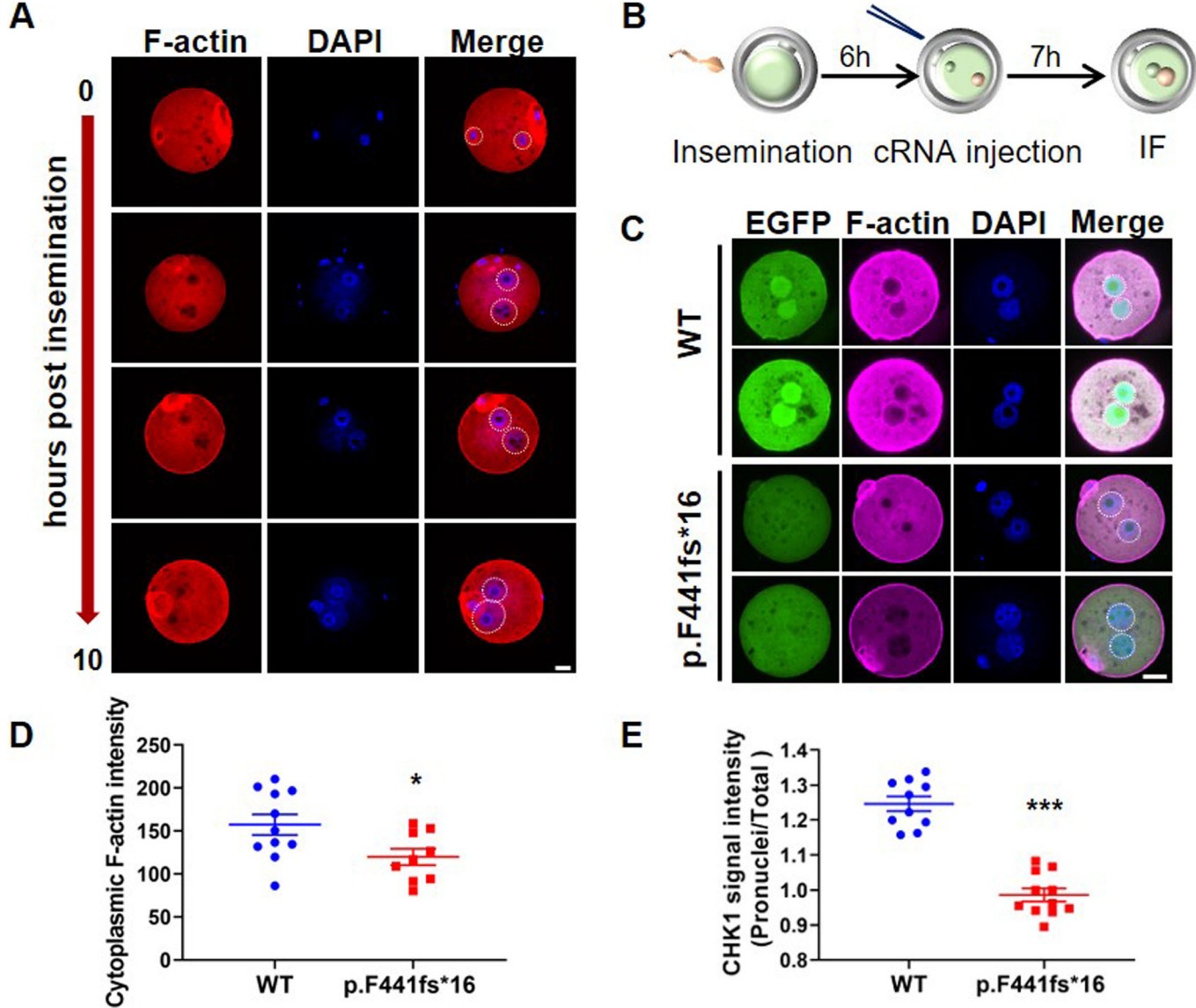

**Figure 2. Disruption of F-actin meshwork by CHK1 mutation in fertilized oocytes.**

(A) Immunofluorescence staining of F-actin with phalloidin and DNA with DAPI, depicting the F-actin meshwork in zygotes at different hours post insemination. Scale bar: 20 μm. The white dashed circle indicates pronuclei. (B) Schematic representation of the cRNA injection procedure in (C). After 6 h of insemination, zygotes with distinct pronuclei were injected with either wild-type or mutant CHK1 cRNA. After 7 h, zygotes in both groups were fixed for immunofluorescence (IF). (C) F-actin fluorescent staining of zygotes overexpressing EGFP-CHK1 (wild-type or p.F441fs*16). The white dashed circle indicates pronuclei. Scale bar: 20 μm. The images shown here were reused in Fig. EV2C for further signal quantification. (D) Quantification of cytoplasmic F-actin signal intensity in wild-type ($n = 11$) or p.F441fs*16 ($n = 9$) zygotes (excluding cortical F-actin) in (C). Two-tailed Student's $t$ tests, *$P < 0.05$. Error bars represent SEM. (E) Quantification of EGFP signal representing CHK1 in pronuclei and total cells separately, followed by comparison of the pronuclei/total cell intensity ratio between the two groups using two-tailed Student's $t$ tests. The total numbers of zygotes counted were 10, 11 for wild-type and p.F441fs*16, respectively. ***$P < 0.001$. Error bars, SEM. About ten zygotes were used for analysis in each group. Source data are available online for this figure.

HEK-293 cells to affirm this interaction. Encouragingly, both the CHK1 antibody and the MICAL3 antibody effectively immuno-precipitated their respective partners, as evidenced by significant bands in the gel (Fig. 4B,C). This outcome was consistent with the CO-IP findings in mouse-fertilized eggs. To offer additional precision in detecting protein–protein interactions, we turned to proximity ligation assay (PLA), a highly specific and sensitive technique for in situ interaction analysis (Soderberg et al, 2006). Employing PLA technology, we examined the in situ interaction

between CHK1 and MICAL3 not only in HEK-293 cells but also in human oocytes and embryos. As anticipated, HEK-293 cells displayed evident red PLA signals, confirming the in situ interaction (Fig. 4D,E; Appendix Fig. S2A). Similarly, distinct PLA signals were also observed primarily in the cytoplasm of the donated human oocyte and early embryos, while control GV oocytes exhibited negligible signals (Fig. 4F,G; Appendix Fig. S2B). These comprehensive results provide robust evidence supporting the interaction between CHK1 and MICAL3.

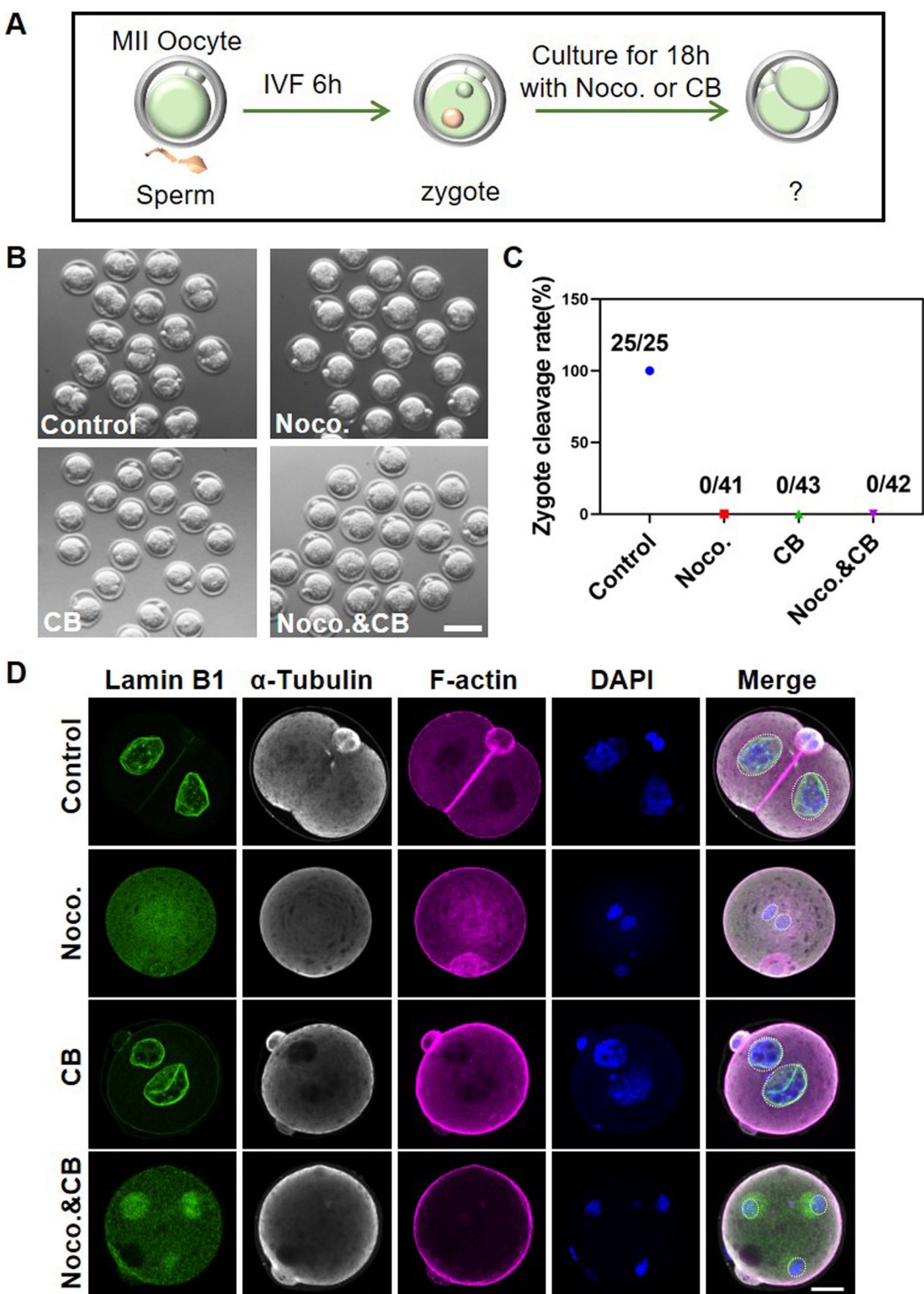

**Figure 3. Inhibiting F-actin affects PNEB of fertilized eggs.**

(A) A diagram depicts the procedure of Nocodazole (Noco.) or Cytochalasin B (CB) treatment. After in vitro fertilization (IVF) for 6 h, the formed zygotes would be cultured with DMSO, 15 µM Noco., 5 µg/ml CB or both of Noco. and CB for 18 h. (B) Representative images show the development results after 18-h treatment. Scale bar: 100 µm. (C) Zygote cleavage rates in each group. The number represents 2-cell embryos/total zygotes. (D) Representative fluorescent staining results of 2-cell embryos or zygotes after 18-h treatment. The white dashed circle indicates chromatin from pronuclei. Lamin B1 (green), α-Tubulin (white), F-actin (red), DNA (blue). Scale bar: 20 µm. The experiment was repeated three times and at least 25 embryos were used for analysis. Source data are available online for this figure.

Consequently, we surmise that CHK1 mutation likely influences F-actin polymerization in fertilized eggs by interacting with MICAL3, thus shedding light on a potential mechanism by which CHK1 contributes to the regulation of F-actin dynamics.

## CHK1 mutation impacts MICAL3's enzyme activity by enhancing their interaction

To explore the intricate mechanism by which CHK1 regulates the F-actin meshwork through its critical target MICAL3, we conducted a thorough investigation into the impact of CHK1 mutation on MICAL3. Initially, we compared the structural alterations induced by CHK1 mutants. Given the absence of a comprehensive crystal structure for CHK1, we applied RoseTTA-Fold to generate both wild-type and mutant CHK1 structures. These models highlighted a shift in the conformation of CHK1's N-terminal and C-terminal domains from a "closed" state to an "open" state following the mutation (Fig. 5A). To validate the predicted changes in the conformation of these models, we employed single-molecule fluorescence resonance energy transfer (smFRET) experiments utilizing reconstituted CHK1 proteins. Specifically, the N-terminal was fused with enhanced green fluorescence protein (EGFP), while the C-terminal was coupled with red fluorescence protein (mCherry). When the proximity between the N-terminal and C-terminal is sufficient, a FRET phenomenon occurs, leading to an enhanced EGFP signal post bleaching of the red mCherry signal (Fig. EV5A,B). Conversely, if the conformation of CHK1 mutant transitions to an "open" state (Fig. 5A), no FRET occurs, resulting in an unaltered EGFP signal following bleaching (Fig. EV5C,D). Intriguingly, both mutations (p.F441fs*16 and p.R379Q) markedly reduced the FRET efficiency in comparison to wild-type CHK1 (Fig. 5B,C). Notably, the sole N-terminal domain of CHK1 (1–266 aa), which served as a negative control, also exhibited the anticipated decrease in FRET efficiency (Fig. 5B,C). Thus, our investigations have substantiated the conformational changes brought about by the mutations, warranting a deeper exploration into the resulting alterations in their biological functions.

Through a protein–protein docking analysis of CHK1 and MICAL3, we observed a shift in their interaction mode post-mutation (Appendix Fig. S3B,C), implying that these structural modifications could impact their interaction with MICAL3. Consequently, we proceeded to individually transfect HEK-293 cells with constructs encoding wild-type and mutant CHK1, followed by conducting CO-IP experiments. Notably, the findings revealed that while CHK1 mutations did not exert any influence on the expression level of MICAL3, they indeed enhanced the interaction between CHK1 and MICAL3 (Fig. 5D).

Given these intriguing findings, a key question emerged: what potential biological effects might occur in MICAL3 as a consequence of this altered interaction? To address this query, we endeavored to detect the enzyme activity of MICAL3. It is widely documented that the oxidation of F-actin by MICAL3 releases $H_2O_2$, leading to F-actin depolymerization and the production of $H_2O_2$ thus serves as a marker for assessing MICAL3 activity (Alto and Terman, 2018; Tominaga et al, 2019). To this end, we assessed $H_2O_2$ production in HEK-293 cells overexpressing either wild-type or mutant CHK1. Notably, the results strikingly indicated a significant increase in $H_2O_2$ production within the mutation groups, as well as in the N-terminal kinase domain group (1–266 aa) (Fig. 5E). Moreover, to further corroborate the altered MICAL3 activity within zygotes, we employed a specific $H_2O_2$ probe and successfully replicated the heightened $H_2O_2$ levels within the mutant group of mouse zygotes (Fig. 5F).

Then, arising from these findings, another question emerged: How does CHK1 precisely activate MICAL3? Initially, we attempted to purify the full-length MICAL3 or its fragments (Appendix Fig. S4). Perhaps due to the fact that MICAL3 is nearly twice the size of MICAL1/MICAL2 (Appendix Fig. S4A) and contains numerous disordered regions (Appendix Fig. S4B), the purification endeavor unfortunately resulted in degradation and yielded very low protein concentrations. Then by reviewing the protein–protein docking models between CHK1s and MICAL3, we can see that wild-type CHK1 mainly interacts with MICAL3 by C-terminal regulatory domain, while the mutated CHK1s (p.F441fs*16 and p.R379Q) with "open conformation" seem to interact with MICAL3 not only by the C-terminal domain but also by the kinase domain (Appendix Fig. S3A–C). The docking models furnish a suggestive insight that CHK1 could potentially engage in the direct activation of MICAL3 through its kinase activity, likely involving a phosphorylation mechanism. Considering there is no antibody specific to phosphorylated MICAL3 and no available purified proteins, we attempted to find an effect assay to confirm the phosphorylation presumption. PF477736 is an ATP-competitive inhibitor of CHK1. It can effectively inhibit the phosphorylation of Cdc2, a downstream target of CHK1, while induces more phosphorylated CHK1 at 345 Serine, indicating the activation of CHK1 (Appendix Fig. S3D). To explore the phosphorylation role of CHK1 in MICAL3 activation, we used CO-IP again to test whether the binding between CHK1 and MICAL3 is affected by PF477736 in HEK-293 cells. After application with PF477736, the interaction between CHK1 and MICAL3 is enhanced (Appendix Fig. S3E). These findings unveil the possibility that the inhibitor could engage in competitive interactions with ATP, potentially triggering compensatory activation of CHK1. This, in turn, may lead to compensatory binding between CHK1 and MICAL3. When considered alongside the docking results (Appendix Fig. S3A–C), it becomes plausible that the active CHK1 mutant may indeed initiate phosphorylation to activate MICAL3.

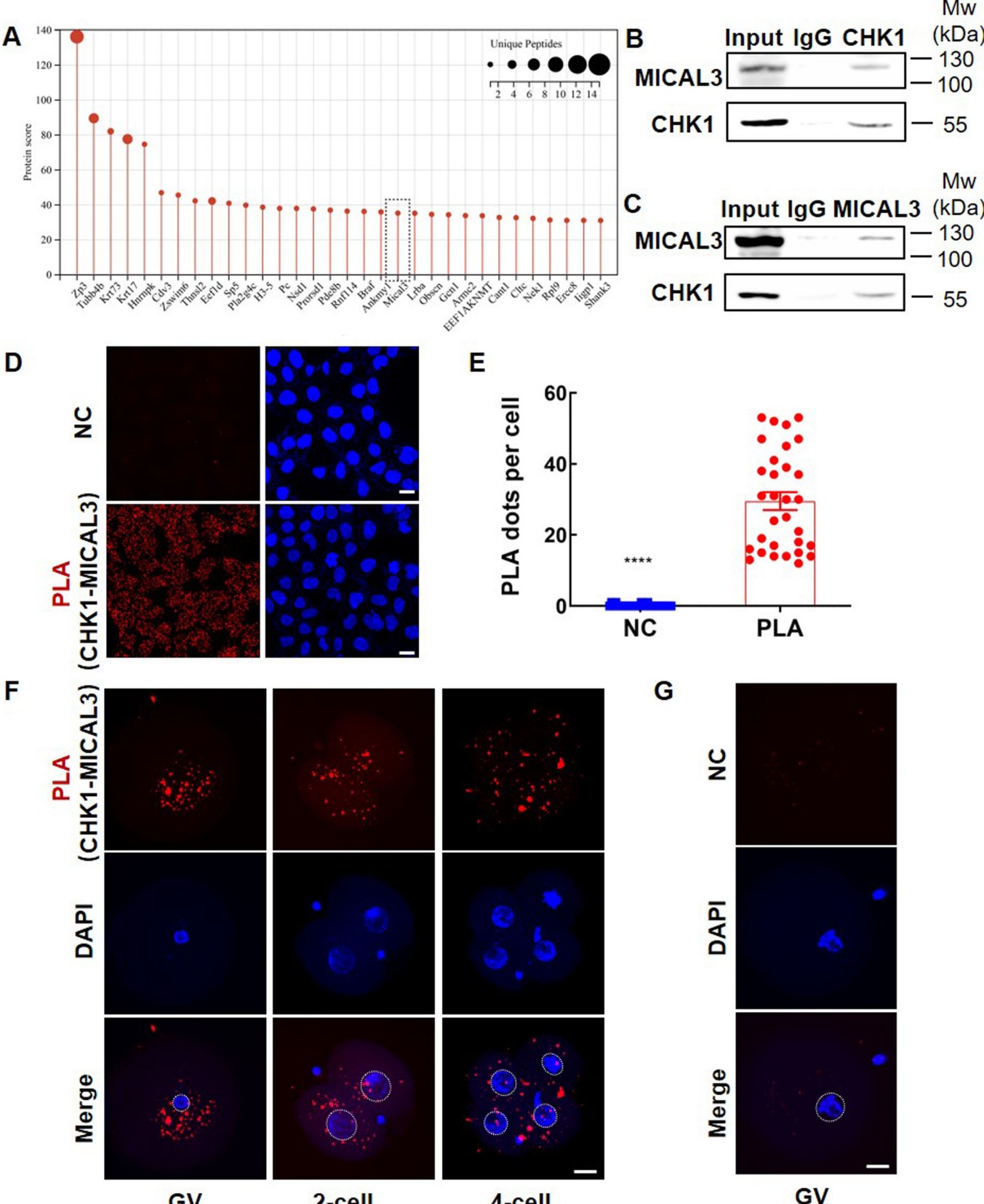

**Figure 4.   CHK1 is able to interact with MICAL3.**

(A) Mass spectrometry result of the CHK1 interactome. Lysates from 6000 mouse zygotes were pooled and used for immunoprecipitation of CHK1. A total of 32 proteins were identified, possibly interacting with CHK1 in zygotes, by subtracting the proteins identified in an IgG control experiment. The unique peptide numbers and protein scores of each protein are displayed. The black dashed rectangle indicates the protein, MICAL3. For more information on the identified protein, please see the source data. (B) CO-IP by the CHK1 antibody shows that CHK1 can precipitate MICAL3. (C) CO-IP by the MICAL3 antibody shows that MICAL3 can precipitate CHK1. (D) Red PLA signals in HEK-293 cells imply that CHK1 can interact with MICAL3 in situ. The PLA analysis was performed three times. Scale bar: 20 μm. (E) Quantification of PLA signals in (D) according to the average red signal dots per cell in each image field and at least 30 images were used for analysis. Two-tailed Student's *t* tests, ****$P < 0.0001$. Error bars, SEM. (F) PLA results of the human GV oocyte, 2-cell embryo and 4-cell embryo display significant red PLA signals. Scale bar: 20 μm. (G) The PLA result of a human GV-stage oocyte without primary antibodies is regarded as normal control (NC). GV, germinal vesicle. Scale bar: 20 μm. The white dashed circle indicates nuclei. Experiments in (B–D) were repeated three times. Source data are available online for this figure.

In summation, these comprehensive findings collectively suggest that CHK1 mutations may expedite the depolymerization rate of F-actin by modulating their "intramolecular self-inhibition" conformation and altering interactions with MICAL3 to enhance its enzymatic activity.

## Inhibiting MICAL3 can effectively improve the cleavage disorder phenotype

To further reinforce and establish the pivotal role of MICAL3 in the processes of PNEB and zygote cleavage, we employed (-)-epigallocatechin gallate, commonly known as EGCG, a component found in green tea known to partially inhibit MICAL monooxygenase activity (Nadella et al, 2005; Terman et al, 2002; Tominaga et al, 2019). We investigated the effects of EGCG on the development of fertilized eggs carrying the CHK1 mutation at various concentrations (Fig. 6A). Encouragingly, at a concentration of 10 μM, EGCG significantly enhanced the cleavage rate of fertilized eggs carrying the CHK1 mutation (p.F441fs*16) (Fig. 6B,D), as well as their subsequent blastocyst development rate (Fig. 6C,E). Interestingly, no significant impact on the blastocyst development rates of zygotes overexpressing wild-type CHK1 was observed for EGCG at any of the concentrations tested (Appendix Fig. S5A,B).

Following the initial differentiation, blastocysts undergo the formation of the inner cell mass (ICM) and trophectoderm (TE), which express cell fate-determining factors such as Oct4 and Cdx2, respectively (Zhang et al, 2018; Zhao et al, 2022). Given the challenges in obtaining blastocysts from fertilized eggs carrying the CHK1 mutation, we further assessed the expression of these two cell fate determinants in blastocysts generated after applying 10 μM EGCG to both the wild-type and mutant groups. The results indicated that the expression levels of Oct3/4 and Cdx2 were comparable between the two groups (Fig. 6F–H). Furthermore, in order to assess whether the inhibition of MICAL3 using the EGCG compound could reverse the $H_2O_2$ and F-actin phenotype, we also investigated the levels of $H_2O_2$ and F-actin in the mutant zygotes following EGCG application. As anticipated, EGCG exhibited the restoration of the F-actin signal, particularly concentrated around the pronuclei (Fig. 6I). Simultaneously, the $H_2O_2$ signal experienced a significant reduction after treatment with EGCG (Fig. 6J).

In addition, another MICAL inhibitor, CCG-1423, has been proven to inhibit the monooxygenase activity of MICAL2(Lundquist et al, 2014), and the molecular docking results indicate a binding pocket not only between the N-termina domain of MICAL3 and EGCG but also between it and CCG-1423 (Appendix Fig. S5C,D). Given the similarity of N-terminal domain between MICAL2 and MICAL3 (Appendix Fig. S4A), we utilized CCG-1423 to further verify the inhibition of EGCG and most of the zygotes

carrying the CHK1 mutation (p.F441fs*16) could successfully develop into blastocyst stage, especially under 1 μM treatment (Appendix Fig. S5E,F).

In summary, these findings provide additional confirmation of the critical involvement of MICAL3 in the CHK1-regulated PNEB process.

## Discussion

Mammalian pronuclear envelope breakdown (PNEB) represents a complex biological event meticulously orchestrated by intricate regulatory mechanisms. Our present investigation sheds light on the pivotal role of CHK1, a critical cell cycle checkpoint kinase, in overseeing and fine-tuning the process of PNEB. In a quiescent state, the wild-type CHK1 adopts a "closed" conformation with a relatively weaker interaction with MICAL3. Concurrently, the activity of MICAL3 remains at a lower level, thus upholding the normal regulatory network of F-actin. This equilibrium ensures the seamless and successful progression of PNEB. However, when a mutation arises in CHK1, disrupting its "closed" conformation, an augmented interaction between CHK1 and MICAL3 occurs. This heightened interaction triggers an elevation in MICAL3's monooxygenase activity, which in turn dismantles the F-actin meshwork, resulting in the impairment of PNEB (Fig. 7). This intricate interplay between CHK1 and MICAL3 underscores their collaborative role in orchestrating the precise execution of PNEB in mammalian fertilized eggs.

As a pivotal kinase in the cell cycle checkpoint, CHK1 participates in a wide array of critical biological processes (Smits and Gillespie, 2015; Zhang and Hunter, 2014). Recent investigations have illuminated its indispensable role in overseeing DNA damage repair during mouse zygote reprogramming (Ladstatter and Tachibana-Konwalski, 2016; Zou, 2016) and in the spindle assembly checkpoint during initial cleavage (Ju et al, 2020). Our recent study has further substantiated its significance in the process of pronuclear envelope breakdown (PNEB) in human-fertilized eggs (Zhang et al, 2021). Patients afflicted with CHK1 mutations solely experience infertility, without manifesting other developmental disorders when they seek medical attention (Gillespie, 2022; Zhang et al, 2021). A long-term follow-up in the future is essential to observe the somatic effects of these CHK1 mutations. Nevertheless, the intricate mechanisms by which CHK1 regulates PNEB remain elusive.

Our laboratory has pioneered a cutting-edge pre-pronuclear transfer (PPNT) technology, facilitating the extraction of female pre-pronuclei shortly after fertilization, without resorting to cytoskeleton inhibitors (Wu et al, 2017). In the present study, we

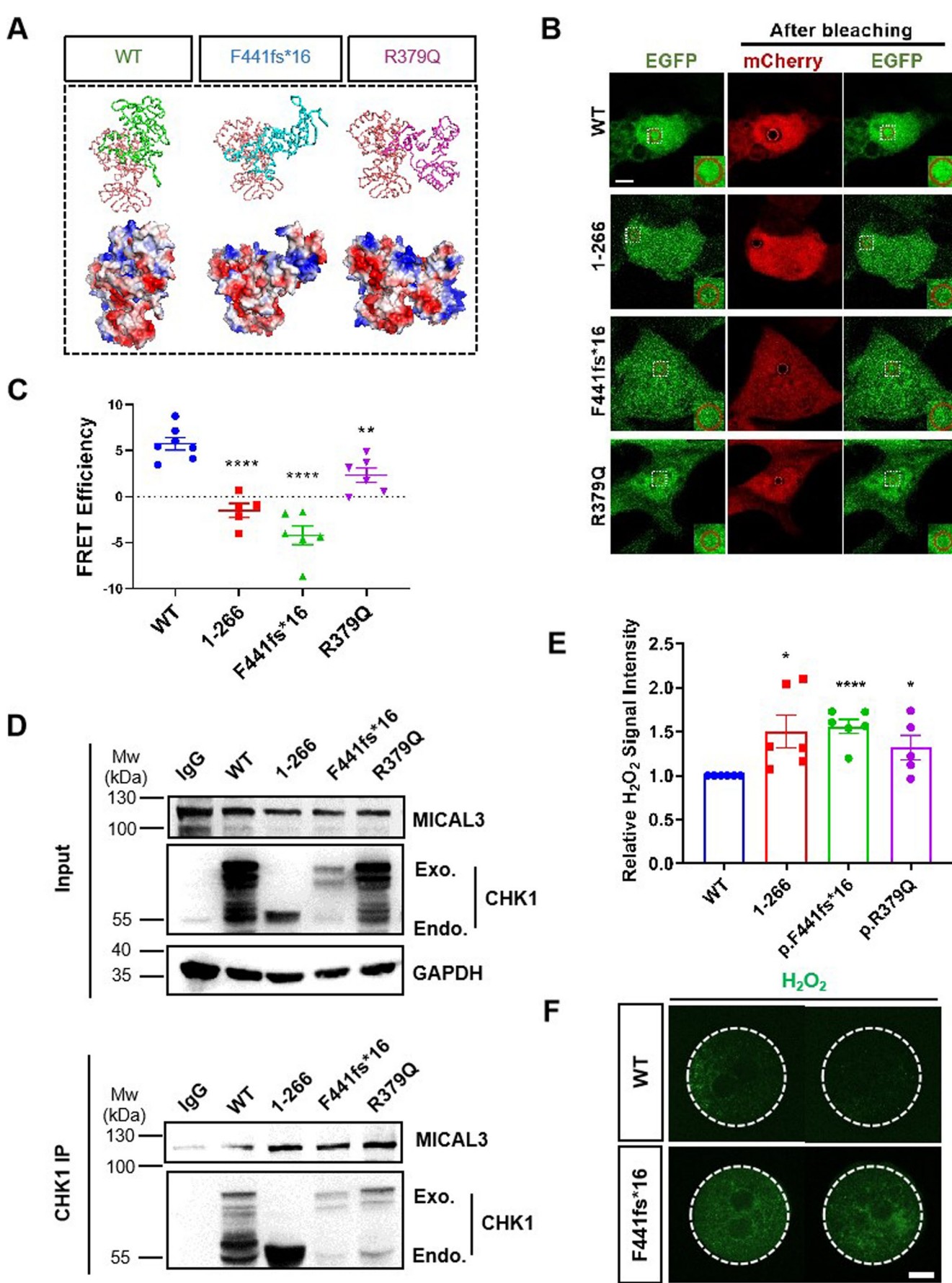

◀  **Figure 5.  CHK1 mutants may affect MICAL3 activity by altering their interaction.**

(A) Structural exhibition of wild-type or mutant CHK1 constructed by RoseTTAFold, revealing altered conformation after mutation. The pink region corresponds to the N-terminal domain, while the green, blue, and purple depict the C-terminal domain of wild-type or mutant CHK1s. (B) Representative images demonstrate the FRET efficiency in different groups, with cropped region of the bleached area for the CHK1. After mCherry bleaching, the EGFP signal significantly enhances in the WT group. The read circles indicate bleaching regions. Scale bar: 20 μm. (C) Fluorescence resonance energy transfer (FRET) efficiency of wild-type (WT) and mutant CHK1. 1–266 represents the first 266 amino acid residues of CHK1, which is the kinase domain serving as a negative control. At least five bleaching fields were used for analysis. Two-tailed Student's $t$ tests, **$P < 0.01$, ****$P < 0.0001$. Error bars, SEM. (D) CO-IP by the CHK1 antibody demonstrates that CHK1 mutation can enhance interaction with MICAL3. (E) Relative $H_2O_2$ signal intensity in different HEK-293 cell groups. The experiment was repeated six times. (F) $H_2O_2$ signal exhibition in zygotes either with wild-type or mutant CHK1, Scale bar: 20 μm. The white dashed circles demonstrate the outline of the zygotes. Source data are available online for this figure.

harnessed the PPNT approach to replace the zygote cytoplasm of a patient harboring a CHK1 mutation, revealing a striking revelation: the arrested zygotes demonstrated the capacity to develop into the blastocyst stage with robust developmental potential. And additional trials through collaboration with reproductive centers is necessary to further assess the benefits of PPNT. Anyway, this intriguing finding suggests that the mutation's adverse impact is primarily centered on the cytoplasm of fertilized oocytes. Moreover, propelled by high-throughput sequencing techniques like whole-exome sequencing (WES), a growing number of pathogenic genes contributing to compromised egg or embryo quality have been successfully pinpointed (Sang et al, 2021). Nevertheless, effective treatment strategies for individuals grappling with such challenges are currently lacking. Exploration of personalized intervention approaches, exemplified by the innovative PPNT technique, holds the promise of ushering in a new era of possibilities for these patients, offering renewed hope for the prospect of conceiving healthy offspring.

The pivotal roles of cytoskeletal components, such as actin, tubulin, and keratin, in oocyte meiosis and early embryo development are widely acknowledged (Clift and Schuh, 2013; Lim et al, 2020). Particularly during the zygote stage, the significance of F-actin in preserving spindle central positioning, facilitating pronuclear migration, and orchestrating the symmetrical division of fertilized eggs has gradually come to light in recent years (Chaigne et al, 2016; Scheffler et al, 2021; Yu et al, 2014). In this context, our study has uncovered a compelling observation: the inhibition of F-actin substantially impedes the rupture of the pronuclear membrane in fertilized eggs. Moreover, in accordance with prior research, our findings indicate that while nocodazole may influence pronuclear migration and fertilized egg quality through microtubule depolymerization (Cavazza et al, 2021; Scheffler et al, 2021), it does not exert a significant impact on pronuclear membrane rupture. These results collectively highlight that although microtubules may play a role in facilitating nuclear membrane rupture in somatic cells (Beaudouin et al, 2002; Margalit et al, 2005; Muhlhausser and Kutay, 2007; Velez-Aguilera et al, 2022), their contribution is outweighed by the pivotal role of F-actin in orchestrating pronuclear envelope breakdown in mammalian zygotes.

As flavoprotein monooxygenase/hydroxylase enzymes, the MICAL family members (MICAL1, MICAL2, and MICAL3) exert their influence by oxidizing the Met44 and Met47 subunits of actin, thereby changing the interactions among actin subunits and culminating in the destabilization and fragmentation of F-actin (Alto and Terman, 2018; Fremont et al, 2017; Kim et al, 2020). Notably, the expression level of MICAL3 in early human embryos surpasses that of MICAL1 and MICAL2 (Yan et al, 2013; Data ref: Yan et al, 2013), suggesting that MICAL3 may possess a more

pronounced role in the initial stages of embryo development. This study has successfully identified and substantiated the interaction between CHK1 and MICAL3 within mammalian zygotes. Importantly, the investigation has revealed that CHK1 gain-of-function mutation amplifies the interaction between CHK1 and MICAL3, consequently heightening the enzymatic activity of MICAL3.

The N-terminal domain of CHK1 interacts with its C-terminal domain to maintain an inactivated "closed" conformation in the quiescent state; When CHK1 undergoes phosphorylation or certain amino acid residues in the conserved motifs of its C-terminal region are mutated, it triggers a transition to an "open" structure, resulting in increased kinase activity (Emptage et al, 2017; Han et al, 2016). Both the predicted CHK1 models and single-molecule fluorescence resonance energy transfer (smFRET) experiments have demonstrated that these mutations disrupt the closed conformation, leading to enhanced kinase activity, which is consistent with our previous report (Zhang et al, 2021). As a protein kinase, gain-of-function CHK1 mutants may influence the enzyme activity of MICAL3 by phosphorylating specific residues through their heightened kinase activity. This hypothesis gains further support from the fact that the N-terminal kinase domain (1–266 aa) group also displayed increased MICAL3 activity. In addition, as suggested by protein–protein docking predictions, the mutated CHK1, in its "open" conformation, may engage its kinase domain more extensively in interactions with MICAL3. Moreover, the observed enhancement of CHK1 interaction with MICAL3 upon CHK1 inhibition could indicate a compensatory mechanism. Taken together, these findings strongly suggest that activated CHK1 may phosphorylate and activate MICAL3. However, the precise underlying modulation mechanism remains to be further elucidated, given the challenges encountered in attempts to purify MICAL3 and the lack of single-cell phospho-proteomics.

In conclusion, this study provides a novel regulatory function of CHK1 in fertilized eggs, and raises a probable explanation for the modulation mechanism of mammalian PNEB. We also raise a feasible and efficient strategy, PPNT, for patients suffering from infertility whose pathogenic genes mainly affect biological issues in the cytoplasm of oocytes or zygotes. In addition, the application of MICAL3 inhibitor supplies a novel intervention for patients suffering from infertility caused by CHK1 mutation.

# Methods

## Pre-pronuclear transfer (PPNT)

After the patient's and donor's oocytes were fertilized by ICSI, the first polar body (PB1) was separated and discarded by mechanical incision of zona pellucida. In all, 3.5–4 h after ICSI, check whether

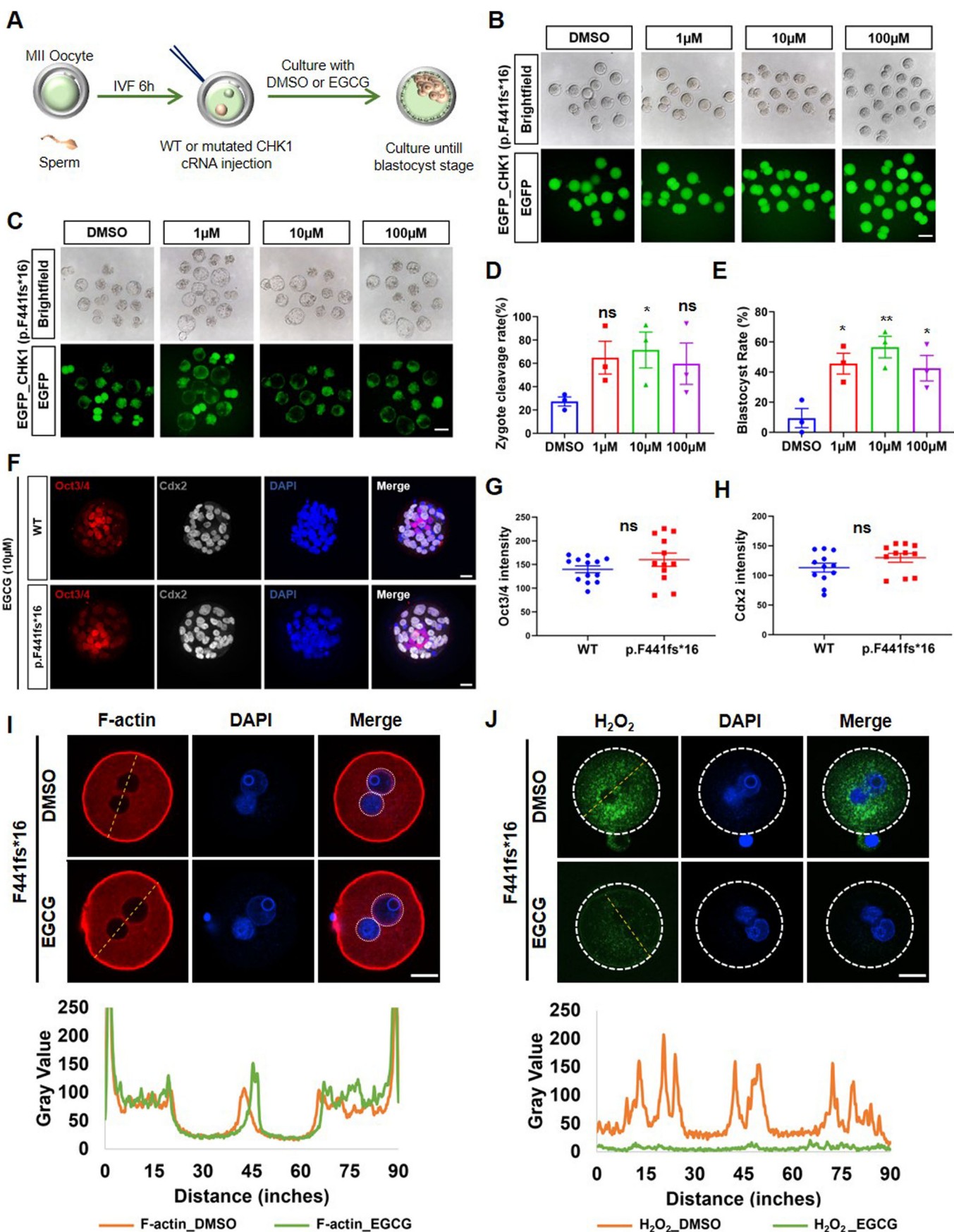

**Figure 6.** **Inhibit MICAL3 by EGCG can efficiently improve the phenotype of zygotes carrying CHK1 mutation.**

(A) A diagrammatic sketch reveals the EGCG treatment procedure. Six hours after in vitro fertilization (IVF), wild-type (WT) or mutated CHK1 cRNA was injected into zygotes which would be cultured with DMSO or EGCG with different concentrations. (B) Representative images reveal the results of 2-cell embryo development at different EGCG concentrations (DMSO/1 µM/10 µM/100 µM). Scale bar: 100 µm. (C) Representative images display the results of blastocyst development at different EGCG concentrations (DMSO/1 µM/10 µM/100 µM). Scale bar: 100 µm. (D) Quantification of zygote cleavage rates in (B). The number of zygotes applied for analysis were 41, 38, 41, 49 in each group (DMSO/1 µM/10 µM/100 µM), respectively. Two-tailed Student's t tests. ns no significant difference. *$P < 0.05$. Error bars, SEM. (E) Quantification of blastocyst rates after EGCG treatment in (C). The number of zygotes applied for analysis were 37, 33, 41, 34 in each group (DMSO/1 µM/10 µM/100 µM), respectively. Two-tailed Student's t tests. *$P < 0.05$, **$P < 0.01$. Error bars, SEM. (F) Blastocysts overexpressing wild-type (WT) or mutated (p.F441fs*16) CHK1 treated with 10 µM EGCG were stained by cell fate-determining factors, Oct3/4 and Cdx2, respectively. Scale bar: 20 µm. (G) Quantification of Oct3/4 signals in the WT ($n = 13$) and p.F441fs*16 ($n = 12$) blastocysts from (F). (H) Quantification of Cdx2 signals in the WT ($n = 12$) and p.F441fs*16 ($n = 11$) blastocysts from (F). Two-tailed Student's t tests. ns no significant difference. Error bars, SEM. (I) Representative images reveal F-actin distribution in zygotes with CHK1 mutation (p.F441fs*16) after DMSO or 10 µM EGCG treatment. The white dashed circle indicates pronuclei. (J) Representative images reveal $H_2O_2$ expression in zygotes with CHK1 mutation (p.F441fs*16) after DMSO or 10 µM EGCG treatment. Scale bar: 20 µm. Line profiles were generated across zygotes below (I, J), encompassing regions that spanned both the pronuclei and cortex. The yellow dashed lines in (I, J) reveal the line profile traces through zygotes under DMSO (orange) or 10 µM EGCG (green) treatment. Source data are available online for this figure.

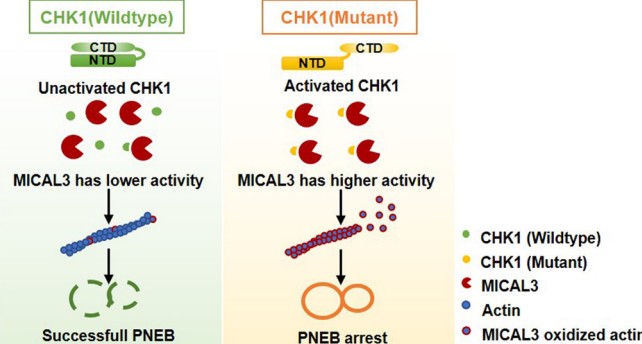

**Figure 7.** **A working model indicating the role of CHK1 in regulating PNEB through interacting with MICAL3.**

In fertilized eggs, the conformation of wild-type CHK1 largely keeps in a "closed" pattern. The interaction between CHK1 and MICAL3 is relatively weaker, which leads to a lower activity of MICAL3. Given the point that MICAL3 contributes to the depolymerization of F-actin, zygotes with lower MICAL3 activity are able to sustain a normal F-actin meshwork to ensure the procedural pronuclear envelope breakdown (PNEB). While the mutant CHK1 keeps in an "open" conformation and enhances the interaction with MICAL3, rendering higher MICAL3 activity. Therefore, the cytoplasmic F-actin meshwork in mutated zygotes is disrupted because of increased MICAL3 activity, leading to PNEB failure.

the patient's mutant second polar body (PB2) is successfully extruded. Then, the adjacent PB2 and PPN were separated and put into G1 medium (Vitrolife) waiting for their complete segregation. PPN was gently isolated from PB2 and then briefly soaked in inactivated Sendai virus (Ishihara Sangyo Kaisha City Ltd). Thereafter, the PPN from the mutant zygote was immediately injected into the perivitelline space of a donor-fertilized egg whose PB1, PPN, and PB2 were removed by the same method. Then the fusion status of the reconstructed zygote was evaluated after 20–30 min. The reconstruction method for donor's PN and patient's cytoplasm is the same as above. All operations were performed on Nikon TE 2000S inverted microscope equipped with 37 °C heating plate (Tokai Hit). All reconstructed zygotes were transferred to G1 medium and cultured in Embryo Scope time-lapse system (Unisense FertiliTech, Denmark). Patients with zygote arrest, as well as healthy control individuals, were recruited in the Center for Reproductive Medicine, Shandong University, China.

All subjects signed informed consents, and this study was reviewed and approved by the Institutional Review Board of Reproductive Medicine, Shandong University. The experiments conformed to the principles set out in the WMA Declaration of Helsinki and the Department of Health and Human Services Belmont Report.

## Copy number variant (CNV) analysis

The 3–4 embryonic stem cells were mechanically isolated for CNV analysis. Whole genome amplification (WGA) was performed by the SurePlex WGA kit (VeriSeq PGS Kit, Illumina), and then the library was prepared and sequenced by high-throughput sequencing platform, DA8600.

## Establishment of human embryonic stem cell (ESC) line

The inner cell mass of a patient's blastocyst was implanted on mitotically inactivated mouse embryonic fibroblasts (MEF) and cultured in human embryonic stem cell medium under 37 °C hypoxic environment (6% $CO_2$, 5% $O_2$). The medium was usually changed every day, and the ESC line grew into clones of appropriate size after six days. The clones were identified by immunofluorescence when they grew to a suitable size.

## Mouse oocyte or embryo collection

Healthy C57BL/6 female mice aged 4–6 weeks were injected with 5 IU pregnant mare's serum gonadotropin (PMSG, NINGBO SANSHENG) followed by 5 IU human chorionic gonadotropin (HCG, NINGBO SANSHENG) after 46–48 h. Sperm were collected in cauda epididymis from C57 male mice aged 8–12 weeks and were capacitated in G-IVF plus medium (Vitrolife) for 1 h. Cumulus oocyte complex (COC) was collected 16 h after HCG injection and the capacitated sperm were added together in G-IVF plus medium covered with mineral oil. After cultured for 4–6 h at 37 °C in 5% $CO_2$ atmosphere, the formed fertilized eggs would be transferred to G1 plus medium (Vitrolife) covered with mineral oil for further research. The animals used in the study were obtained from Beijing Vital River Laboratory Animal Technology Co., Ltd. and were cultured in SPF animal center of Shandong University. The protocol for the animal study was reviewed and approved by the Institutional Review Board of Reproductive Medicine, Shandong University.

## In vitro synthesis and microinjection of cRNAs

The plasmids were linearized with appropriate restriction endonuclease. According to the manufacturer's method, 5 'capped mRNA was synthesized with mMESSAGE mMACHINE T7 Transcription Kit (Invitrogen, AM1344), and then poly (A) was added using Poly(A) Tailing Kit (Invitrogen, AM1350). The synthesized cRNAs were purified with RNeasy MinElute Cleanup Kit (Qiagen, 74204) and then dissolved in nuclease-free water. About 5 pl of cRNA solution (1400 ng/UL) was microinjected into the plasma of fertilized eggs.

## Immunofluorescence

Embryos were fixed in 4% paraformaldehyde (Solarbio) for 30 min and permeabilized in PBS containing 0.3% TritonX-100 for 20 min. After blocking with 1% bovine serum albumin (Sigma) in PBS for 1 h, they were incubated with the corresponding primary antibodies diluted with blocking solution at 4 °C overnight. After washing several times, they would be incubated with the secondary antibodies for 60 min the next day, and then counterstained with 4′,6-diamidino-2-phenylindole (DAPI, Vector Laboratories) for 10 min. After sealing, they would be imaged with a confocal laser-scanning microscope (Andor, Dragonfly) under a ×40 objective lens. The primary antibodies are shown in Appendix Table S2.

For embryonic stem cells or HEK-293(MeilunBio, PWE-HU007) cells: Rinse the culture dish containing embryonic stem cells or cells gently with warm PBS once, then fix it at room temperature with 4% paraformaldehyde for 20 min. After washing it with cold PBS for three times, permeate it with PBS containing 0.3% TritonX-100 for 20 min. Then block it with PBS containing 5% bovine serum albumin for 1 h, and incubate it with primary antibody diluted with blocking solution at 4 °C overnight. The following procedures were the same as embryo immunofluorescence. The primary antibodies are listed in Appendix Table S2.

## Co-immunoprecipitation (CO-IP) experiments

Mouse-fertilized eggs or HEK-293 cells were collected by protein lysis buffer (ThermoFisher, 78501) containing protein phosphatase inhibitor (Beyotime, P1046). In total, 1 µg CHK1 antibody and corresponding 1 µg IgG antibody were separately added and incubated overnight at 4 °C. Prepare Protein A/G Mix Magnetic Beads (Merck, LSKMAGAG10) and mix gently the next day. Take 1.5-ml centrifuge tubes and add 50-µl magnetic beads into each tube. Place the centrifuge tube on the magnetic stand and discard the supernatant storage solution. Add 500 µl PBST (0.1% Tween 20 in PBS) and shake violently for 10 s to wash the beads. Then add the protein lysis incubated overnight into centrifuge tubes with magnetic beads, and roll at room temperature for 30 min. Following that, clean the magnetic beads with 500 µl PBST for three times. After the last cleaning, add 40 µl mixture of pre-prepared RIPA and loading buffer in each tube, and denature them at 95 °C for 10 min. After that, put them on the magnetic stand and collect the supernatant for further analysis after denaturation.

For mass spectrum analysis of the collected mouse zygotes, the supernatant was carried out by SDS-polyacrylamide gel electrophoresis. When the sample was running into the separation gel about 1.5 cm, the SDS-polyacrylamide gel would be stained by Coomassie Blue Staining Solution (Beyotime, P0017F). Then the stained gel would be collected for mass spectrometry, performed by PTM Biolabs, Inc.

For western blot analysis of HEK-293 cells, the supernatant was separated by SDS-polyacrylamide gel electrophoresis and transferred to PVDF membrane (Millipore). After the membrane was incubated with primary antibodies at 4 °C overnight, it would be incubated by HRP conjugated second antibodies for 1 h at room temperature, and then was developed by Image Lab gel imaging system (Bio-Rad). The primary antibodies used are shown in Appendix Table S2.

## Cell transfection

HEK-293 cells were cultured in DMEM/High glucose medium (HyClone, SH30243.01B) together with 10% fetal bovine serum (BI, 04-001-1ACS) at 37 °C with 5% $CO_2$. After the cell density reached at a 70–80% fusion degree, the plasmids would be transfected with Lipofectamine 3000 Transfection Kit (Invitrogen, L3000015) according to the instructions given by the manufacturer.

## Proximity ligation assay

Proximity ligation assay (PLA) is a special immunofluorescence assay, which is used to analyze the interaction between proteins in situ. We used Duolink® In Situ Red Starter Kit Mouse/Rabbit (Sigma, DUO92101-1KT) here to perform in situ detection of protein–protein interactions in cells or embryos according to the manufacturer's instructions. After fixation, permeabilization and blocking pretreatment, the cells or embryos were incubated with primary antibodies (rabbit anti-CHK1 antibody and mouse anti-MICAL3 antibody) overnight. The primary antibodies were not added to the negative control. The PLUS and MINUS PLA probes were used as secondary antibodies. The red detection reagent was used to indicate protein interaction signals, which were then collected by confocal laser microscope for Z-scan across the nucleus. and analyzed by ImageJ. The quantification of PLA signals in HEK-293 cells relied on determining the average number of red dots per cell in each image field. To be more specific, the total PLA red dots were divided by the number of nuclei, as indicated by DAPI staining.

## Single-molecular fluorescence resonance energy transfer (smFRET)

The CHK1 constructs were fused with enhanced green fluorescence protein (EGFP) in the N-terminal and red mCherry protein in the C-terminal, whose cDNA sequences were confirmed by Sanger sequencing in Sangon Biotech. HEK-293 cells were then transfected with either wild-type or mutant CHK1. Two-proton laser-scanning microscopy (LSM880 NLO) was used to perform the experiment. For the FRET efficiency calculations, the intensity alterations for EGFP-CHK1 in the circled FRET region were divided by the EGFP-CHK1 signal before FRET.

## Detection of $H_2O_2$

In HEK-293 cells, the Amplex® red hydrogen peroxide/peroxidase assay kit (Invitrogen, A22188) was applied to measure the

production of $H_2O_2$ according to the manufacturer's scheme. The reaction was carried out in a volume of 100 μl, including 50 μM Amplex® red reagent, 0.1 U/ml HRP reaction buffer, and 200 μM NADPH. Then 20 μl cell culture supernatant was added and incubated at room temperature for 30 min. The fluorescence signal was collected under the conditions of 560-nm excitation light and 590-nm emission light. In zygotes, the ROSGreen™ Hydrogen peroxide probe (MX5202) was applied following the manufacturer's instructions.

### Data analysis

GraphPad Prism 8.0 was used for statistical analysis. Most experiments were repeated at least three times. Non paired *t* tests were used for the comparison between two groups. The significance of GraphPad was evaluated as follows: $*P < 0.05$, $**P < 0.01$, $***P < 0.001$, $****P < 0.0001$. The quantification of F-actin and EGFP-CHK1 signal was performed by ImageJ and subsequently optimized by Microsoft Excel.

## Data availability

This study includes no data deposited in external repositories.

The source data of this paper are collected in the following database record: biostudies:S-SCDT-10_1038-S44319-024-00267-7.

## Peer review information

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

## Acknowledgements

The authors sincerely thank the patients recruited in the Center for Reproductive Medicine, Shandong University, China for their participation and selfless donation. We are grateful to Drs. Yuan Gao, Ming Gao, Rusong Zhao, Chuanxin Zhang, Jingzhu Song from Shandong University, China for their enthusiastic technical advice. We thank Translational Medicine Core Facility of Shandong University for consultation and instrumentavailability that supported this work. This study was supported by the National Natural Science Foundation of China (82192874, 82071606, 82171842, 32170817); Basic Science Center Program of NSFC (31988101); CAMS Innovation Fund for Medical Sciences (2021-I2M-5-001); Shandong Provincial Key Research and Development Program (2020ZLYS02); Taishan Scholars Program of Shandong Province (ts20190988, tsqn201909194); the National Key Research and Development Program of China (2021YFC2700400); Postdoctoral Research Fund of Gusu School in Nanjing Medical University (GSBSHKY202303); Fundamental Research Funds of Shandong University (2023QNTDO04), Innovative research team of high-level local universities in Shanghai: SHSMU-ZLCX20210201.

## Author contributions

**Honghui Zhang**: Conceptualization; Resources; Data curation; Software; Formal analysis; Supervision; Validation; Visualization; Writing—original draft; Writing—review and editing. **Ying Cui**: Validation; Writing—review and editing. **Bohan Yang**: Methodology; Writing—review and editing. **Zhenzhen Hou**: Data curation. **Mengge Zhang**: Data curation. **Wei Su**: Data curation. **Tailai Chen**: Data curation. **Yuehong Bian**: Data curation; Supervision. **Mei Li**: Data curation; Funding acquisition. **Zi-Jiang Chen**: Supervision; Funding acquisition. **Han Zhao**: Supervision; Funding acquisition. **Shigang Zhao**: Supervision; Funding acquisition; Writing—original draft; Writing—review and editing. **keliang wu**: Supervision; Funding acquisition; Writing—original draft; Writing—review and editing.

Source data underlying figure panels in this paper may have individual authorship assigned. Where available, figure panel/source data authorship is listed in the following database record: biostudies:S-SCDT-10_1038-S44319-024-00267-7.

## Disclosure and competing interests statement

The authors declare no competing interests.

# Expanded View Figures

**Figure EV1.  Replacement of the mutant cytoplasm by pre-pronuclear transfer (PPNT).**

(A) Time-lapse images reveal the embryos on the first day (Day 1) and the third day (Day 3) after fertilization. The control embryo without mutation is in 2-cell stage on Day 1 and in 8-cell stage on Day 3. While the embryos from two patients separately carrying CHK1 mutations p.R379Q and p.F441fs*16 are still in zygote stage with distinct pronuclei on Day 3. The white arrowheads indicate pronuclei. (B) Time-lapse images captured on Day 6 depict the reconstructed embryos. The upper panel displays embryos composed of the patient's female pronucleus (PN) and the donor's enucleated cytoplasm (CP), while the lower panel shows embryos with the donor's pronucleus (PN) and the patient's cytoplasm (CP). (C) A diagram illustrates the process of pre-pronuclear transfer (PPNT) conducted between zygotes obtained from a patient with CHK1 mutation (p.R379Q) and a healthy donor. Mature oocytes (MII) are collected from both the patient and the donor, and then subjected to intracytoplasmic sperm injection (ICSI) for fertilization. The first polar bodies (PB1) formed after fertilization are removed. After the second polar bodies (PB2) have been naturally extruded, the female pre-pronuclei (PPN) along with PB2 are isolated and dissociated from both the patient's and donor's zygotes. The PPN from the patient's zygote is transferred into the perivitelline space (PVS) of an enucleated oocyte from the donor. This reconstructed oocyte develops into a zygote containing two pronuclei. Simultaneously, the female PPN from the donor's zygote is injected into the patient's oocyte PVS using the same method. This leads to the formation of a zygote with two pronuclei in the patient's oocyte. (D) Quantitative analysis of pluripotency marker signals between the reconstructed stem cell line (ESC_Em-1) and the stem cell line (ESC_PF-1) treated with the inhibitor in (E). At least three representative images were employed for the quantitation. Error bars, SEM. (E) Representative images showing pluripotency markers of an embryonic stem cell line derived from the patient's blastocyst after treatment with a CHK1 inhibitor. Scale bar: 100 μm. (F) A diagram indicates the application of ICM (inner cell mass) and TE (trophoblast) of the treated blastocysts. (G) Sanger sequencing chromatograms of Em-1 in both forward and reverse sequencing direction demonstrate wild-type (WT) genotype.

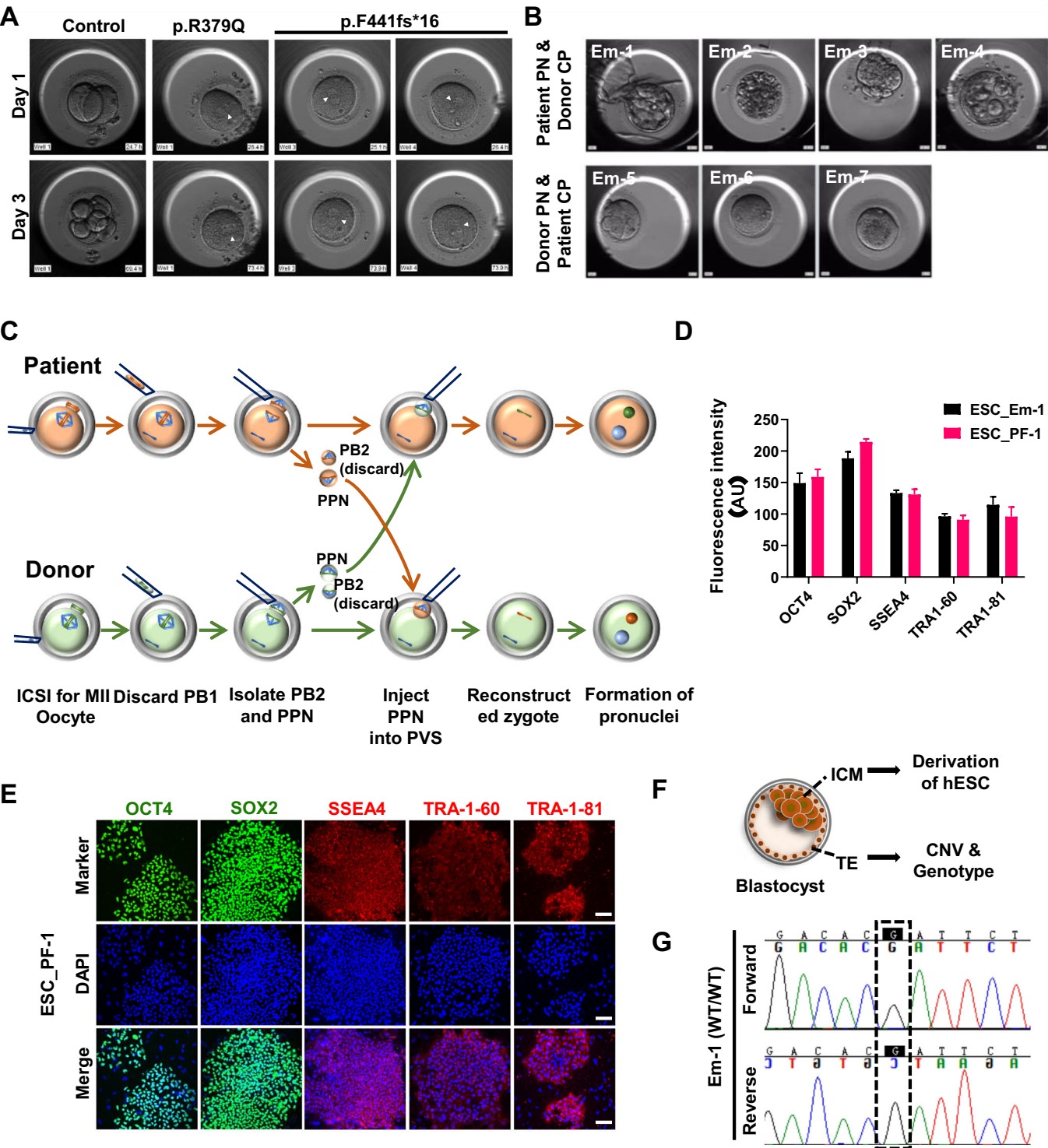

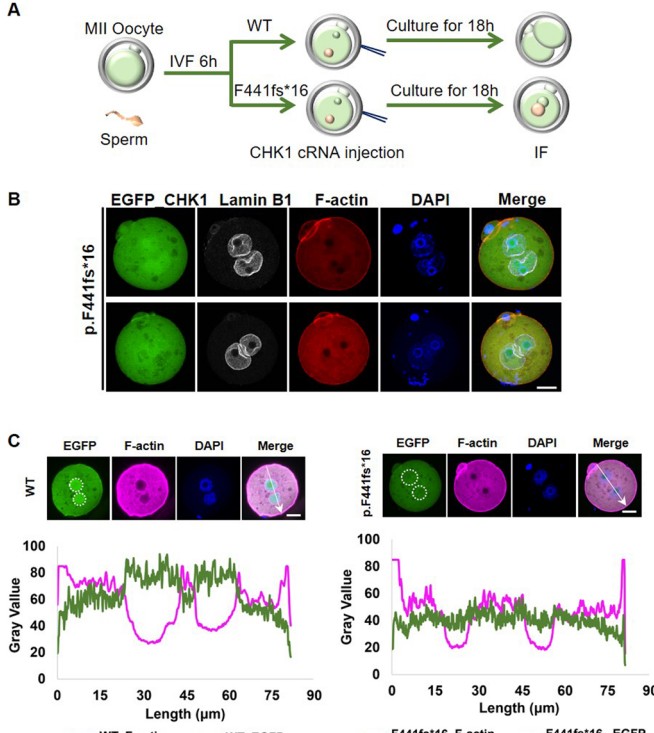

**Figure EV2.  Zygotes overexpressing mutant CHK1 (p.F441fs*16) arrest with clear pronuclear member.**

(A) A diagram reveals the cRNA injection procedure in zygotes. Six hours after in vitro fertilization (IVF), the zygotes with clear pronuclei were injected with wild-type (WT) or mutated (p.F441fs*16) CHK1 cRNA, followed by being cultured for 18 h to perform immunofluorescence (IF) staining. (B) The IF result shows that the arrested zygotes with CHK1 mutation (p.F441fs*16) have obvious pronuclear member indicated by the Lamin B1 antibody. Scale bar: 20 μm. (C) Line profiles were generated across zygotes, encompassing regions that spanned both the pronuclei and cortex. These profiles vividly illustrate the spatial distribution of F-actin and EGFP signal in both wild-type and mutant zygotes. White dashed circles in zygotes indicate the pronuclei, while the white arrowheads reveal the line profiles below. Scale bar: 20 μm. The images used in Fig. EV2C are derived from Fig. 2C for the signal quantification.

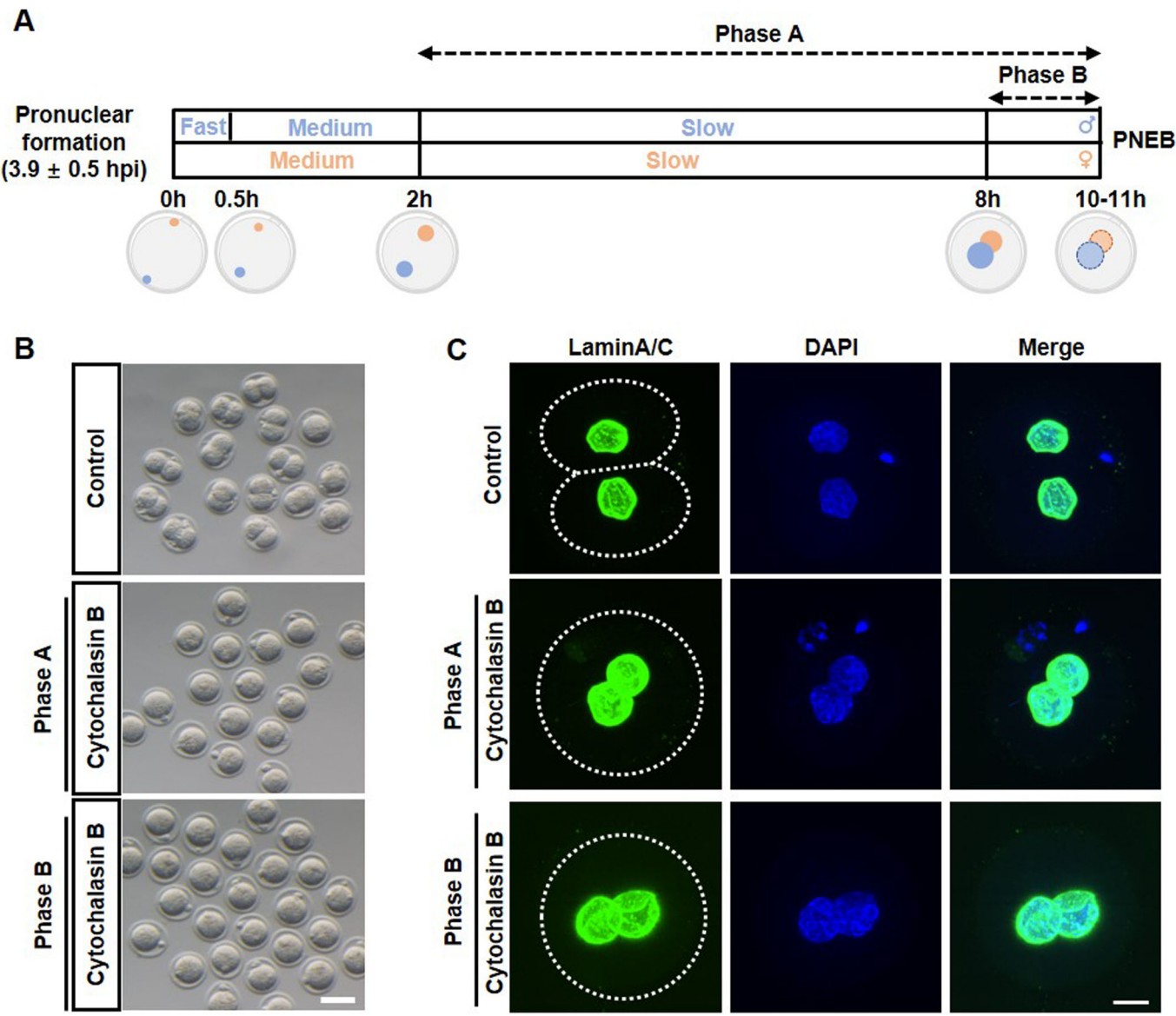

**Figure EV3. Inhibiting F-actin can disturb the PNEB event.**

(A) The provided diagram outlines critical time points subsequent to pronuclei formation, which typically occurs around 3.9 ± 0.5 h post insemination (hpi). Notably, the male pronucleus experiences fast, medium and slow migration phases, while the female pronucleus undergoes medium and slow migration stages, culminating in pronuclear envelope breakdown (PNEB). (B) Captured images depict the developmental outcomes of zygotes across various groups subjected to cytochalasin B treatment during phase A and phase B. The scale bar corresponds to 100 μm. (C) Immunofluorescence results from the embryos depicted in (B) are presented, showcasing specific markers. Scale bar: 20 μm.

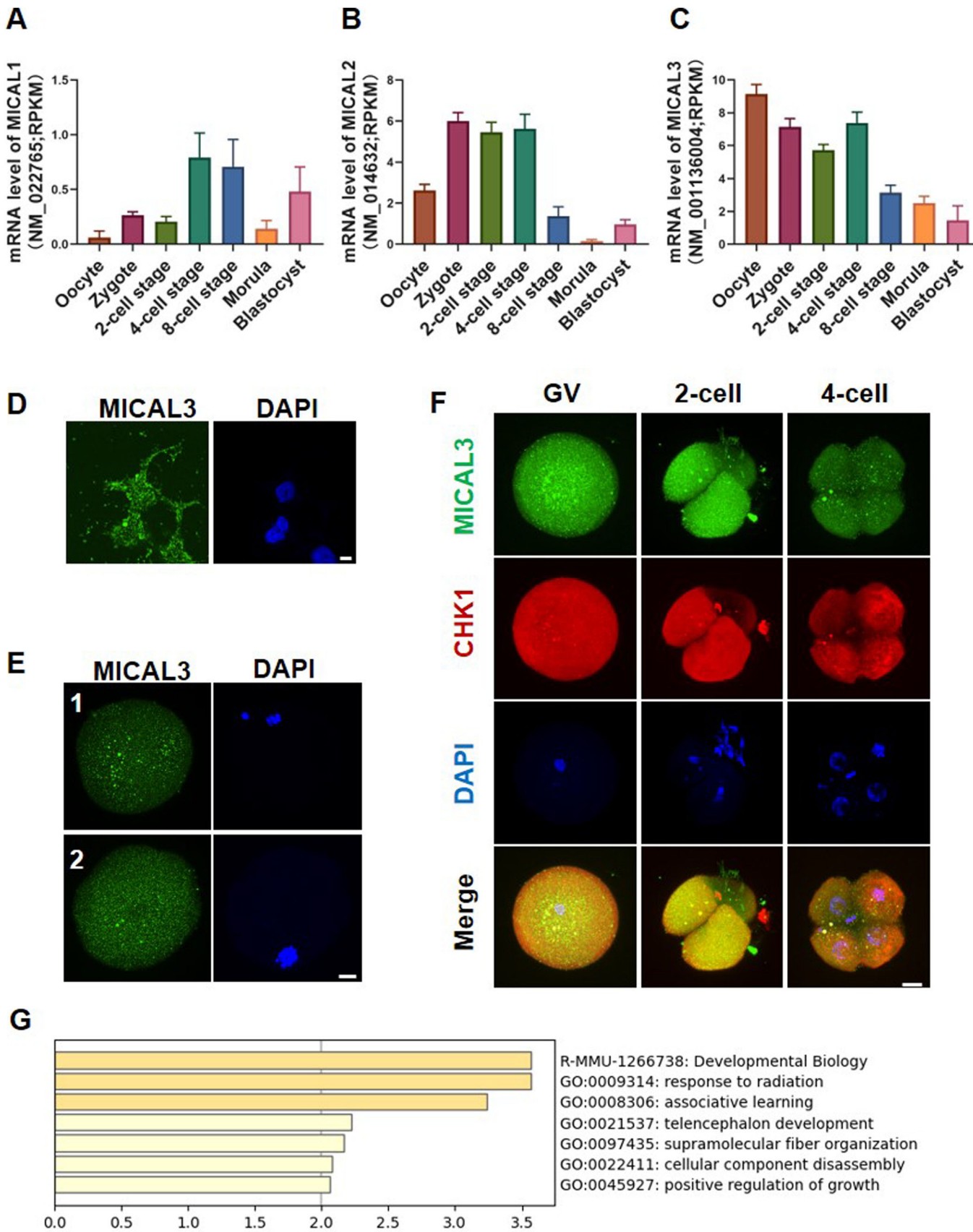

**Figure EV4. The expression and localization of human MICAL3 protein.**

(A–C) The mRNA expression levels of MICAL family members (MICAL1/MICAL2/MICAL3) in human oocytes and preimplantation embryos according to the published transcriptome data (Yan et al, 2013; Data ref: Yan et al, 2013). The data included at least three oocyte or embryo replication in each stage. Error bars, SEM. (D) The localization of MICAL3 in HEK-293 cells. Scale bars: 20 μm. (E) Human oocytes immunofluorescence staining result shows that MICAL3 mainly localizes in cytoplasm. Scale bars: 20 μm. (F) Immunofluorescence staining of both MICAL3 and CHK1 in the human GV oocyte, 2-cell and 4-cell embryos. Of note, the oocyte and embryos were reused after completing proximity ligation assay and the experiments in (E, F) were performed two times because of the rarity of human oocytes. Scale bars: 20 μm. (G) Bar graph of enriched terms across proteins identified as potentially interacting with CHK1 in Fig. 4A, colored by P values and analyzed by Metascape.

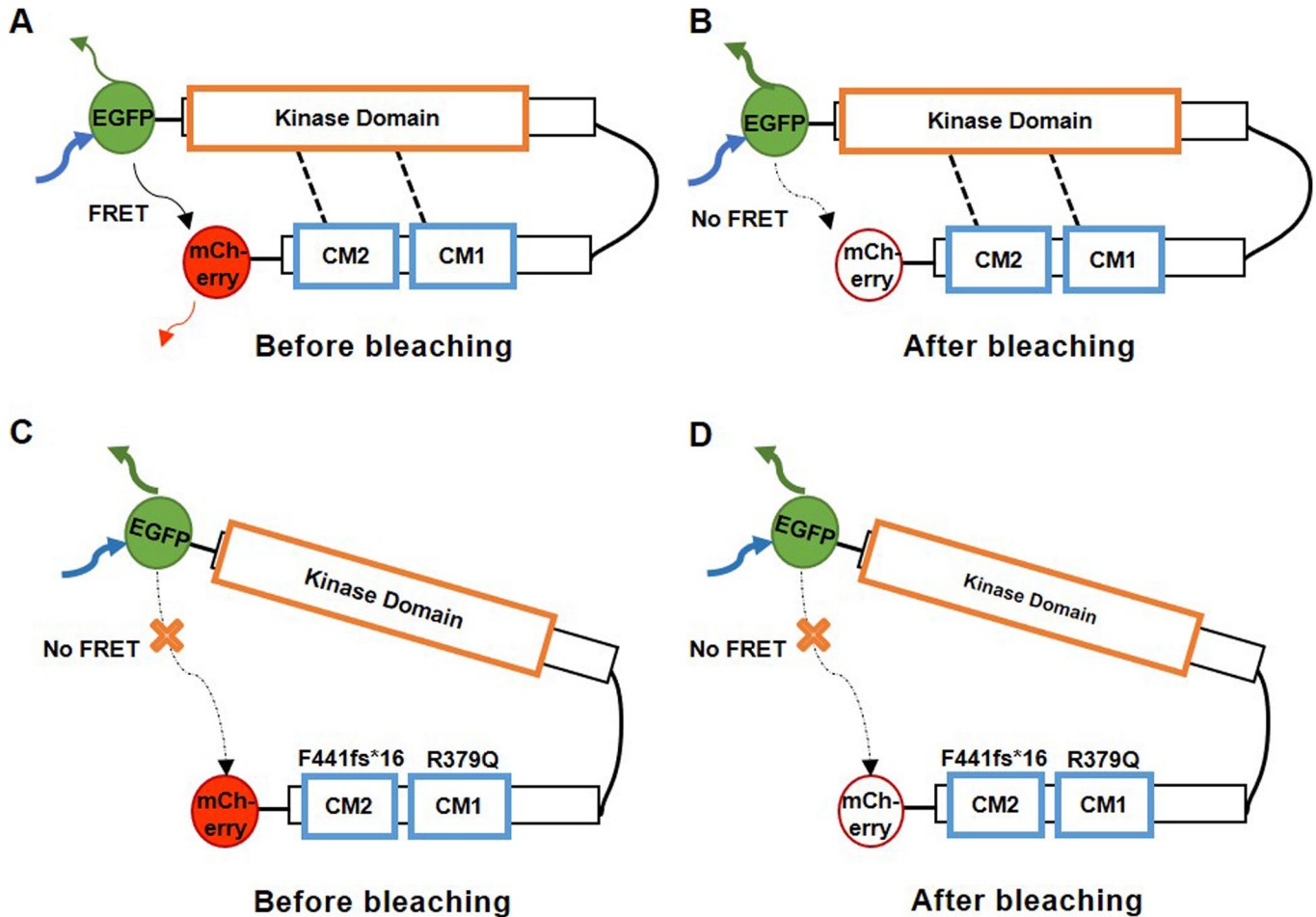

**Figure EV5. Diagrams depict the principle of single-molecular fluorescence resonance energy transfer (smFRET).**

The CHK1 protein composed of a N-terminal kinase domain and a C-terminal domain with two conserved motifs (CM1 and CM2). For smFRET, CHK1 protein was fused with enhanced green fluorescence protein (EGFP) and red mCherry protein separately in the two terminals. (A) Wild-type CHK1 keeps in a "closed" state with interaction between the N-terminal and C-terminal domains. When the two terminal domains are closed enough, the EGFP signal is excited and some of its emission light will be transferred to excite mCherry signal, resulting in reduced EGFP emission signal. (B) After bleaching mCherry, the EGFP emission signal can't be transferred to mCherry and thus will be enhanced. (C) The N-terminal and C-terminal domains of mutant CHK1 hold an "open" conformation and their distance is not that closed to generate FRET. (D) Even though bleaching mCherry, the EGFP emission signal is not disturbed. The blue curves with arrowhead indicate excitation light of EGFP. Green curves with arrowhead show emission light of EGFP. The black curve with arrowhead denotes part of the EGFP emission signal, indicating the happen of smFRET. The dotted black lines with arrowhead indicate no FRET phenomenon.

