## [Peer Review File · EMBO Reports]

CHK1 Controls Zygote Pronuclear Envelope Breakdown by Regulating F-actin through Interacting with MICAL3

Honghui Zhang, Ying Cui, Bohan Yang, Zhenzhen Hou, Mengge Zhang, Wei Su, Tailai Chen, Yuehong Bian, Mei Li, Zi-Jiang Chen, Han Zhao, Shi-Gang Zhao, and keliang wu

Corresponding author(s): *keliang wu (wukeliang@sdu.edu.cn)* , *Shi-Gang Zhao (zsg0108@sdu.edu.cn)*

Review Timeline:

Submission Date:	4th Oct 23
Editorial Decision:	22nd Nov 23
Revision Received:	8th Feb 24
Editorial Decision:	14th Mar 24
Revision Received:	8th Jul 24
Editorial Decision:	26th Aug 24
Revision Received:	3rd Sep 24
Accepted:	5th Sep 24

Editor: Deniz Senyilmaz Tiebe

Transaction Report:

Dear Dr. Wu,

Thank you for the submission of your research manuscript to our journal, which was now seen by three referees, whose reports are copied below.

My apologies for this unusual delay in getting back to you. It took longer than anticipated to receive the full set of referee reports.

Referees express interest in the proposed role of CHK1 in regulation of Zygote Pronuclear Envelope Breakdown. However, they also raise significant concerns that need to be addressed to consider publication here.

Given these positive recommendations, we would like to invite you to submit a revised manuscript. Please revise your manuscript with the understanding that the referee concerns (as in their reports) must be fully addressed and their suggestions taken on board. Please address all referee concerns in a complete point-by-point response. Acceptance of the manuscript will depend on a positive outcome of a second round of review. It is EMBO reports policy to allow a single round of major experimental revision only and acceptance or rejection of the manuscript will therefore depend on the completeness of your responses included in the next, final version of the manuscript.

We realize that it is difficult to revise to a specific deadline. In the interest of protecting the conceptual advance provided by the work, we recommend a revision within 3 months. Please discuss the revision progress ahead of this time with me if you require more time to complete the revisions, or if you have questions or comments regarding the revision (also by video chat).

1. A data availability section providing access to data deposited in public databases is missing (where applicable).
2. Your manuscript contains statistics and error bars based on $n=2$. Please use scatter plots in these cases.

You can submit the revision either as a Scientific Report or as a Research Article. For Scientific Reports, the revised manuscript can contain up to 5 main figures and 5 Expanded View figures, and it should not exceed 27000 characters. If the revision leads to a manuscript with more than 5 main figures it will be published as a Research Article. In this case the Results and Discussion section should be separate. If a Scientific Report is submitted, these sections have to be combined. This will help to shorten the manuscript text by eliminating some redundancy that is inevitable when discussing the same experiments twice. In either case, all materials and methods should be included in the main manuscript file.

4) a .docx formatted letter INCLUDING the reviewers' reports and your detailed point-by-point responses to their comments. As part of the EMBO publication's Transparent Editorial Process, EMBO reports publishes online a Review Process File (RPF) to accompany accepted manuscripts. This File will be published in conjunction with your paper and will include the referee reports, your point-by-point response and all pertinent correspondence relating to the manuscript. <https://www.embopress.org/page/journal/14693178/authorguide#transparentprocess>

5) a complete author checklist, which you can download from our author guidelines <https://www.embopress.org/page/journal/14693178/authorguide>. Please insert information in the checklist that is also reflected in the manuscript. The completed author checklist will also be part of the RPF.

6) Please note that all corresponding authors are required to supply an ORCID ID for their name upon submission of a revised manuscript (<<https://orcid.org/>>). Please find instructions on how to link your ORCID ID to your account in our manuscript tracking system in our Author guidelines <<https://www.embopress.org/page/journal/14693178/authorguide#authorshipguidelines>>

Additional information on source data and instruction on how to label the files are available: <https://www.embopress.org/page/journal/14693178/authorguide#sourcedata>

9) Our journal encourages inclusion of *data citations in the reference list* to directly cite datasets that were re-used and obtained from public databases. Data citations in the article text are distinct from normal bibliographical citations and should directly link to the database records from which the data can be accessed. In the main text, data citations are formatted as follows: "Data ref: Smith et al, 2001" or "Data ref: NCBI Sequence Read Archive PRJNA342805, 2017". In the Reference list, data citations must be labeled with "[DATASET]". A data reference must provide the database name, accession number/identifiers and a resolvable link to the landing page from which the data can be accessed at the end of the reference. Further instructions are available at <http://www.embopress.org/page/journal/14693178/authorguide#referencesformat>

- the name of the statistical test used to generate error bars and P values,
- the number (n) of independent experiments (please specify technical or biological replicates) underlying each data point,
- the nature of the bars and error bars (s.d., s.e.m.),
- If the data are obtained from n Program fragment delivered error `Can't locate object method "less" via package "than" (perhaps you forgot to load "than"?) at //ejpvfs23/sites23b/embor_www/letters/embor_decision_revise_and_review.txt line 56.' 2, use scatter blots showing the individual data points.

12) Please also note our reference format:

I look forward to seeing a revised version of your manuscript when it is ready. Please let me know if you have questions or comments regarding the revision.

Kind regards,

Deniz Senyilmaz Tiebe

Deniz Senyilmaz Tiebe, PhD
Scientific Editor
EMBO Reports

Referee #1:

CHK1 Controls Zygote Pronuclear Envelope Breakdown by Regulating F-actin through Interacting with MICAL3, HongHui Zhang et al.

In this manuscript by HongHui Zhang et al., the authors explored the role of CHK1 in the process of nuclear envelope breakdown in zygotes (PNEB) describing a mechanism involving the regulation of MICAL3 activity and downstream modulation of F-actin meshwork stability. They also showed that in human embryos with mutation of CHK1, PNEB may not occur and as a result, zygotes arrest and do not progress further in development. The authors suggest that this is due to increased cytoplasmic localization of the mutated CHK1 protein and its increased interaction with MICAL3, which results in MICAL3 activation by CHK1. They show that this leads to destabilization of the F-actin meshwork and prevents PNEB. They also show that this phenotype can be rescued by transfer of the female chromatin into enucleated fertilized eggs obtained from healthy donors. Overall, I find the study interesting and valuable. Especially, because it proposes new mechanism of CHK1 involvement in PNEB and suggests novel approaches for infertility treatment for patients with CHK1 mutations.

However, there are points that should be clarified or addressed experimentally before the work can be published:

1. Authors claim that CHK1 mutation does not have any effect on somatic cells and that its phenotype is observed exclusively in zygotes. They show that after human pronuclear transfer, an ESC line derived from one rescued human embryo could be characterized by normal expression of pluripotency markers. However, at the same time, they show MICAL3 and CHK1 interactions in HEK-293 cells as a part of this work suggesting that the mechanism of NEB in somatic cells might involve these factors. Since authors suggest that their method of rescuing embryos with the CHK1 mutation could have potential application in the clinic, it is important to explore whether there is truly a negligible effect on somatic cells:

- CHK1 function links to other master regulators of the cell cycle, and CHK1 is known to be engaged in many other important processes. Thus, I would expect to learn more about CHK1 and the presence and function of MICAL3 in somatic cells and the potential effects of mutated CHK1 in such cells. I understand that after pre-pronuclear transfer, rescued and developing embryos might express both the WT CHK1 allele (originating from sperm) and the potentially mutated CHK1 allele (originating from the transferred female chromatin). However, since CHK1 function described by the authors shows that this mutation might have a dominant effect and mutated CHK1 might destabilize F-actin even in presence of the WT form, I think it is crucial to provide detailed information on its potential effect in somatic cells. This is especially the case, because, the two publications that authors cite to support their claim, do not provide extensive analysis to prove that CHK1 mutants do not have any effects in somatic cells (Gillespie, 2022; Zhang et al., 2021).

- Additionally, did the authors check if the mutated CHK1 was present in the ESC line derived from rescued blastocyst (at the protein and/or RNA level)? Could they provide additional information on other quality parameters of this ESC line compared to WT line (rather than the line obtained after CHK1 inhibitor treatment)? For example: do cells proliferate normally? What is the distribution of the cell cycle phases? Is the proportion of G2/M cells as would be expected? Is cell death occurring at the expected rate?

2. Sample size: Out of 4 human embryos with the CHK1 mutation used in the study, undergoing pre-pronuclear transfer, two were rescued (Em-1 and Em-4) and progressed to the blastocyst stage, and two (EM-2 and EM-3) did not reach blastocyst stage and degenerated having 2-6 cells, according to Table S1. There were only 3 control embryos with chromatin transferred to mutant zygotes. Only one ESC line was derived from the rescued embryos to test its quality (in a limited way as stated above)

- Since the authors propose new method for potential treatment of the patients carrying CHK1 mutations and make the following statement in the abstract, I would expect them to have tested more embryos to show the efficiency of the method.

"(...)In this study, we conducted experiments where pre-pronuclei from zygotes with CHK1 mutations were transferred into the cytoplasm of normal enucleated fertilized eggs. Remarkably, this approach successfully rescued the zygote arrest caused by the

mutations, resulting in the production of high-quality blastocysts. (...)"

3. (lines 149-157):

- The difference in F-actin quantity is not very clear in the pictures (Fig. 2C,D). I suggest moving Suppl. Fig. 2C to the main text to underline the difference around the pronuclei rather than in the cytoplasm. Additionally, the Methods section lacks information on methods of quantification.

- Authors claim that mutant CHK1 zygotes can be characterized by an increased fraction of CHK1 in the cytoplasm and they support this by citing Fig. 2C and 2E whereas these panels do not show this.

4. (Line 212-216 and 232-236): referring to Fig. 4 and Suppl. Fig. 4F, the authors show CHK1 and MICAL3 PLA and co-localization results in GV oocytes, as well as 2-cell and 4-cell embryos. Is there any particular reason why these analyses were not completed in zygotes specifically? I would assume that it is most important to show this stage from the point of view of this manuscript.

5. (Line 202): I would find it informative to explore more the topic of 32 proteins identified as potentially interacting with CHK1. I think that grouping them additionally by function or engagement in known processes would be very insightful.

6. In the introduction section, authors state that "it is common to observe embryos arresting at the two-pronuclei (2PN) stage (Rawe et al, 2003)".

- In the cited paper the rate of the 2PN arrest is not given. However, according to Zhang et al., 2021, the rate of arrest at the zygote stage is 2%. Could the authors provide an actual number with a reference rather than this general statement?

- How frequent is mutation of CHK1 in patients?

7. (Line 176): Authors say that nocodazole treatment "Intriguingly" resulted in zygotes progressing to prometaphase with PNEB occurrence. This effect is not so much intriguing in my opinion as it has been known for many years both in somatic cells and embryos. Also, in Discussion section (lines 415-417), the authors claim that their study uncovered that not microtubules but actin is essential for (P)NEB which I find a little overstated, at least for involvement of microtubules.

8. (Line 187) There should be a reference to Fig. S3B when "Phase B" is mentioned. Otherwise, I find it confusing.

9. Figure 2A, 2C and other Figures: at least the polar body and chromatin should be marked in the pictures

10. Figure 6I and 6J: quantifications have not been provided.

Referee #2:

Zhang and colleagues hand in their work entitled "CHK1 controls Zygote Pronuclear Envelope Breakdown by Regulating F-Actin through Interacting with MICAL3".

The authors start out in intact human oocytes and determine pronuclear envelope breakdown (PNEB) as a key event indicating the start of zygote development in control/wt oocytes as compared to oocytes facing gain-of-function mutants of CHK1 kinase. Indeed, these mutants inhibit progression beyond the zygote stage. The action of the mutant is initially characterized in a remarkable pre-pronuclear transfer experiment, in which exchange of the ooplasm rescues development, suggesting that it is an early-stage cytoplasmic action of CHK1 that inhibits progression of zygote development. The authors then move one step forward and show that PNEB is likewise inhibited when compromising actin polymerization (not MT). The obvious hypothesis is then tested: is there a link between CHK1 activity and regulation of actin assembly. Identification of CHK1 interaction partners addresses this. Among several interactors, MICAL3 is picked as a candidate to explain a link between CHK1 activity and actin assembly in the zygote. Interaction studies in zygotes and somatic cells confirm the initial identification, which seems to be increased in mutant variants of CHK1. Finally, specific inhibition of MICAL3 function rescues zygote development suggesting that its inhibition by lowering CHK1 activity is required for PNEB and zygote development.

Taken together, this is a very nice piece of work and foresee publication in EMBO reports given that one central aspect will be addressed adequately. In the current version of the manuscript, the link between CHK1 action and the actin cytoskeleton. I appreciate that interaction between CHK1 and MICAL3 is documented and that inhibition of MICAL3 by EGCG rescues at an optimal concentration. However, the focus on MICAL3 is not well documented. Mass spec data come "out of the blue", i.e. there are no background control values, the number of unique peptides is low as well as the overall score. Other, more abundantly identified proteins are not further commented on. This needs to be better documented/clarified. Moreover, even though the outcome of the EGCG inhibition seems obvious, its specificity is neither commented nor tested at all. How do we know that what we see is specific? There should be an alternative approach with e.g. overexpression of negative (mono-oxygenase dead) mutant variants or knockdown to reduce activity of endogenous MICAL3 using an alternative to EGCG.

Finally, the statement of statistics are often superficial; how many independent experiments were performed and what is the actual sample size is not consistently given.

Of minor importance are numerous grammatical and spelling mistakes that should and will be corrected prior to publication.

Referee #3:

In the manuscript EMBOR-2023-58263V1 with the title "CHK1 Controls Zygote Pronuclear Envelope Breakdown by Regulating F-actin through Interacting with MICAL3" the authors reveal a novel function for CHK1 in controlling MICAL3 monooxygenase activity for F-actin disassembly necessary for successful Pronuclear Envelope Breakdown (PNEB) and subsequent first mitosis. In general, the authors use a variety of common techniques such as confocal microscopy, smFRET, immunoblotting, Co-IPs and in-vitro assays next to pre-pronuclear transfer (PPN) as uncommon but required technique to uncover mechanistic insights in the process of PNEB in fertilized zygotes from humans and mice as well as in the derived embryonic stem cell lines or human HEK cells. As the use of PPN has to be considered questionable in an ethical background it allows for new experimental approaches to gain insights into the developmental processes and subsequently develop future therapeutic options. The authors show remarkable development of functional blastocysts out of patients' zygotes with severe defects in embryogenesis due to CHK1 mutations prohibiting first mitotic division after fertilization. Using confocal imaging the authors show a disruption of F-actin structures around the pronuclei presents in the two-pronuclei (2PN) stage of the embryos of patients with mutant CHK1. Further experiments show actin-dependency rather than microtubule involvement mediating PNEB using specific drugs for either cytoskeletal component. Mass spectrometry analysis of CHK1 in mouse zygotes revealed the actin disassembly factor MICAL3. Proximity ligation assays in HEK cells as well as in zygotes confirmed association between CHK1 and MICAL3. Characterization of mutant CHK1 displayed an open conformation compared to wt CHK1 going hand in hand with an increased association with MICAL3 in immunoprecipitation experiments. Increased enzymatic activity of MICAL3 by mutant CHK1 leads to higher F-actin depolymerization rates. Finally, the authors partially rescue their phenotype by mutant CHK1 by inhibiting MICAL3 monooxygenase function with epigallocatechin gallate (EGCG), a MICAL monooxygenase activity inhibitor. Overall, the authors took a great effort in putting together this data.

In summary, I support the publication of the manuscript, if the following points can be addressed by the authors:

Image display in figures:

We recommend changing the displayed colors in figures in a color-blind version to avoid green and red images in the same figure. Easiest here would be to change red to magenta.

Major points:

1) In Fig. 2 the authors say "As anticipated, zygotes carrying the CHK1 mutation (p.F441fs16) displayed a significant reduction in F-actin signal, particularly in cytoplasm and around pronuclei, compared to wild-type zygotes (Figure 2C, 2D and Figure S2C). Additionally, consistent with prior findings (Zhang et al., 2021), the expression of mutant CHK1 in the pronuclei markedly diminished, with an increased proportion exhibiting cytoplasmic localization (Figure 2C, 2E and Figure S2C)."

I do appreciate the authors quantification results, however the lower image of wt zygotes in Fig. 2C does not show an increased CHK1 localization in the two indicated pronuclei compared to mutant CHK1 and to me the cytoplasmic CHK1 wt localization compared to the image below that for mutant CHK1 do look roughly the same. To better represent their findings the authors should choose a different representative image for the lower wt panel showing CHK1 localization in the pronuclei. The linescans in Fig. S2 across the whole cell covering both of the pronuclei and plot the fluorescence intensity for CHK1-EGFP and Phalloidin provide additional useful data that could be displayed in the main figure. In general, showing the increased localization of CHK1 in the pronuclei together with an increase in F-actin fluorescence around the pronuclei and decrease in the cytoplasmic intensity in the main figure with representative images would be more suitable.

2) The authors performed a PLA assay in Fig. 4 for an interaction-readout between CHK1 and MICAL3, however there are some general controls missing for the cell-based PLA. As indicated in the materials and methods the authors stained without the primary antibodies as negative control in Fig. 4D. Additionally, it would be good to see PLA assays in HEK cells with the following conditions: primary antibodies individually with either secondary antibody to exclude unspecific binding of the secondary antibody to the other primary antibody, a positive control of a known CHK1 interactor (see Blasius et al. 2011) and a knockdown of either CHK1 or MICAL3 via siRNA compared to non-targeted siRNA over three independent biological replicates to further support their findings on the interaction of CHK1 and MICAL3. In this connection, Fig. 4E shows the quantification of PLA dots per cell without cellular labeling for the cell body. It is unclear how the authors distinguished between different cells, as it looks that the quantification represents the dots of each cell in the provided representative image in Fig. 4D. There is no indication of biological replicates for this experiment as the authors wrote "Most experiments were repeated at least three times", please clarify. Please also explain the ImageJ quantification of PLA dots in the materials and methods section in more detail as there is no co-staining of a cellular marker besides DAPI for nuclear staining. Are these maximum intensity projections?

if so, the authors should indicate.

3) The authors perform smFRET measurements of mutant CHK1 compared to wt CHK1 and plot FRET efficiency in Fig. 5B and C. The figure would benefit from a crop of the bleached region for the CHK1 panel for the reader to better see the fluorescence increase after successful FRET for wt CHK1. Additionally, there is not enough information about the calculations of the FRET efficiency in the materials and methods section.

4) In Fig. 5 the authors performed assays with pyrene labelled actin to quantify the depolymerization rates of CHK1 mutants compared to wt CHK1. Similar to other methods described in this paper I have some things to point out here next to the lack of information in the methods section. The authors use the commercially available kit from Cytoskeleton for their experiments but end up measuring only 1 timepoint for a kinetic assay during 1 h incubation. It would be essential to read out multiple times during their 1 h incubation time to determine exact kinetic curves for actin depolymerization in three biological replicates, as this assay is sensitive enough to provide this information (see Mu et al. 2020 PMID: 31871199). Furthermore, the author should plot their readouts similar to the paper I refereed to in the last sentence and add one control to the experiment, such as the depolymerization of F-actin alone in G-Buffer compared to the addition of wt and mutant CHK1.

5) The authors' aim to rescue the mutant phenotype by applying the EGCG, an inhibitor for MICAL monooxygenase activity and claim: "As anticipated, EGCG exhibited a partial restoration of the F-actin signal, particularly concentrated around the pronuclei (Figure 6I)." It is quite difficult to visually see the partial restoration of the F-actin signal around the pronuclei. A linescan analysis similar to Fig. S2 would better support their claims. In addition, applying the EGCG drug to an actin depolymerization assay showing less depolymerization of F-actin or performing the PLA assay in combination with EGCG would strengthen the authors claims and make their rescue approach more valid. If this is not possible the authors can discuss the partial rescue in more detail.

Minor points:

6) In Fig. 4 B and C the authors should indicate the molecular weight. Furthermore, if they can the authors should add a loading control to Western blot analysis, as the amount of pulled down protein levels are fairly low compared to the input thereby displaying a weak interaction/association under basal and closed conformation conditions. How do the authors explain the significant PLA dot increase of CHK1 and MICAL3 in their cell-based PLA assay compared to the weak interaction in their Co-IPs under basal conditions?

7) Fig. 5D is missing the molecular weight indication and a loading control for CHK1 IP blot (right panel) is essential here if the authors want to claim an "enhancement" of mutant CHK1 and MICAL3 interaction/association compared to wt CHK1. If the GAPDH loading control of the input panel also refers to the CHK1 IP panel the authors can neglect this point and provide an image of the whole membrane in their supplementary information for transparency reasons.

RE: EMBOR-2023-58263V1

Fab. 12, 2024

Dr. Deniz Senyilmaz Tiebe

Scientific Editor

EMBO Reports

Dear Editor and Reviewers,

We extend our sincere gratitude for the invaluable comments and constructive suggestions provided for our manuscript titled "CHK1 Controls Zygote Pronuclear Envelope Breakdown by Regulating F-actin through Interacting with MICAL3 (EMBOR-2023-58263V1)." Additionally, we deeply appreciate the detailed modification strategies that have proven immensely beneficial in enhancing the manuscript's quality.

We have diligently addressed each of the points raised by the reviewers, and we present a summary of our responses below. Comments from reviewers are in blue, while comments from us (authors) are in black. We genuinely hope that our efforts adequately meet your expectations and requirements.

Reviewers' comments:

Referee #1:

In this manuscript by HongHui Zhang et al., the authors explored the role of CHK1 in the process of nuclear envelope breakdown in zygotes (PNEB) describing a mechanism involving the regulation of MICAL3 activity and downstream modulation of F-actin meshwork stability. They also showed that in human embryos with mutation of CHK1, PNEB may not occur and as a result, zygotes arrest and do not progress further in development. The authors suggest that this is due to increased cytoplasmic localization of the mutated CHK1 protein and its increased interaction with MICAL3, which results in MICAL3 activation by CHK1. They show that this leads to

destabilization of the F-actin meshwork and prevents PNEB. They also show that this phenotype can be rescued by transfer of the female chromatin into enucleated fertilized eggs obtained from healthy donors. Overall, I find the study interesting and valuable. Especially, because it proposes new mechanism of CHK1 involvement in PNEB and suggests novel approaches for infertility treatment for patients with CHK1 mutations.

However, there are points that should be clarified or addressed experimentally before the work can be published:

Re:

We sincerely appreciate your valuable insights and thoughtful suggestions, which provide significant guidance for our research. After a thorough review of your comments and recommendations, we have incorporated additional discussions and explanations that we believe will address your concerns. We kindly request you to review the detailed responses provided below and hope they meet with your approval.

1. Authors claim that CHK1 mutation does not have any effect on somatic cells and that its phenotype is observed exclusively in zygotes. They show that after human pronuclear transfer, an ESC line derived from one rescued human embryo could be characterized by normal expression of pluripotency markers. However, at the same time, they show MICAL3 and CHK1 interactions in HEK-293 cells as a part of this work suggesting that the mechanism of NEB in somatic cells might involve these factors. Since authors suggest that their method of rescuing embryos with the CHK1 mutation could have potential application in the clinic, it is important to explore whether there is truly a negligible effect on somatic cells:

- CHK1 function links to other master regulators of the cell cycle, and CHK1 is known to be engaged in many other important processes. Thus, I would expect to learn more about CHK1 and the presence and function of MICAL3 in somatic cells and the potential effects of mutated CHK1 in such cells. I understand that after pre-pronuclear transfer, rescued and developing embryos might express both the WT CHK1 allele

(originating from sperm) and the potentially mutated CHK1 allele (originating from the transferred female chromatin). However, since CHK1 function described by the authors shows that this mutation might have a dominant effect and mutated CHK1 might destabilize F-actin even in presence of the WT form, I think it is crucial to provide detailed information on its potential effect in somatic cells. This is especially the case, because, the two publications that authors cite to support their claim, do not provide extensive analysis to prove that CHK1 mutants do not have any effects in somatic cells (Gillespie, 2022; Zhang et al., 2021).

Re:

Thank you for your insightful comments and suggestions. We have thoroughly considered your comments and implemented necessary revisions to address the raised concerns.

We recognize the significance of investigating CHK1 function in somatic cells, especially given its pivotal role as a master regulator of the cell cycle. As illustrated in Figure 5, overexpressed CHK1 mutations alter the interaction with MICAL3, increase MICAL3's enzyme activity, and induce an elevated depolymerization rate of F-actin in HEK-293 cells. However, the mechanism of nuclear envelope breakdown (NEB) in somatic cells (Beaudouin *et al*, 2002; Mühlhäusser & Kutay, 2007), mainly involving microtubules rather than F-actin, may differ from that in zygotes. Consequently, the altered MICAL3 activity and F-actin depolymerization with mutated CHK1 in somatic cells may have minimal impact on NEB. Patients carrying CHK1 mutations exhibit infertility with zygote cleavage disorder but lack other common diseases, suggesting a hyper-sensitivity of the G2 arrest mechanism in fertilized zygotes, with a distinct role for CHK1 in zygotes compared to somatic cells (Gillespie, 2022; Zhang et al., 2021).

CHK1 has been identified as being overexpressed in various human tumors, including liver (Hong *et al*, 2012), cervix (Xu *et al*, 2013), breast (Verlinden *et al*, 2007) and nasopharyngeal carcinoma (Sriuranpong *et al*, 2004). Additionally, transgenic mouse embryonic fibroblasts with an extra copy of CHK1 undergo malignant

transformation, likely due to the restriction of oncogene-induced replication stress (López-Contreras *et al*, 2012). By impeding the progression of the cell cycle, CHK1 affords cells adequate time to respond to DNA damage factors, thereby promoting cell survival. This adaptation allows cells with elevated CHK1 expression to thrive in tumor microenvironments characterized by substantial replication stress, fostering the development of malignant tumors. Consequently, concerning the impact on somatic cells, gain-of-function CHK1 mutations may potentially be associated with tumor occurrence, necessitating extended follow-up studies for patients and their families. In the event of a potential tumor-inducing effect, we propose a strategic approach by combining pre-pronuclear transfer technology with preimplantation genetic diagnosis could offer a strategy to filter embryos without CHK1 mutation, addressing potential concerns about tumor induction.

- Additionally, did the authors check if the mutated CHK1 was present in the ESC line derived from rescued blastocyst (at the protein and/or RNA level)? Could they provide additional information on other quality parameters of this ESC line compared to WT line (rather than the line obtained after CHK1 inhibitor treatment)? For example: do cells proliferate normally? What is the distribution of the cell cycle phases? Is the proportion of G2/M cells as would be expected? Is cell death occurring at the expected rate?

Re:

Regarding the presence of the mutated CHK1 in the ESC line derived from the rescued blastocyst, we have conducted Sanger sequencing of the reconstructed blastocyst (Em-1) to ascertain the genotype of the ESC line. The findings reveal that the ESC line does not harbor the mutated CHK1 allele (see Response Figure 1, corresponding to Figure EV1F, G in the manuscript), and these results have been incorporated into the revised manuscript (Page 6, Line 133-135). Certainly, it is crucial to assess the impact of mutated CHK1 on the quality parameters of the ESC line, including cell proliferation, cell cycle phases, and cell death. We acknowledge the importance of such considerations. Perhaps, insights from future follow-ups on

patients with CHK1 mutations could shed light on somatic cell effects. We emphasize the need for increased attention to patients with CHK1 mutations (Page 17, Line 368-373).

We hope that these revisions comprehensively address your concerns and enhance the robustness and clarity of our manuscript. Your feedback has been invaluable, and we welcome any additional suggestions you may have.

Response Figure 1. Sanger sequencing of the reconstructed blastocyst. (A) A diagram demonstrates the analyses applied in the reconstructed blastocyst. The genotype was identified by Sanger sequencing of TE. ICM, inner cell mass; TE, trophoblast; hESC, human embryonic stem cell; CNV, copy number variation. (B) Chromatograms from Sanger sequencing of Em-1, obtained through both forward and reverse sequencing. Em-1, Embryo 1.

2. Sample size: Out of 4 human embryos with the CHK1 mutation used in the study, undergoing pre-pronuclear transfer, two were rescued (Em-1 and Em-4) and progressed to the blastocyst stage, and two (EM-2 and EM-3) did not reach blastocyst stage and degenerated having 2-6 cells, according to Table S1. There were only 3 control embryos with chromatin transferred to mutant zygotes. Only one ESC line was derived from the rescued embryos to test its quality (in a limited way as stated above)

- Since the authors propose new method for potential treatment of the patients carrying CHK1 mutations and make the following statement in the abstract, I would expect them to have tested more embryos to show the efficiency of the method.

"(...)In this study, we conducted experiments where pre-pronuclei from zygotes with CHK1 mutations were transferred into the cytoplasm of normal enucleated fertilized eggs. Remarkably, this approach successfully rescued the zygote arrest caused by the mutations, resulting in the production of high-quality blastocysts. (...)"

Re:

Thank you for bringing attention to the sample size in our study and for raising important points regarding the potential implications of our findings. We have carefully considered your comments and made the following adjustments to address the concerns:

We understand your concerns regarding the limited number of embryos analyzed and its potential impact on the overall significance of the study. Unfortunately, due to the scarcity and precious nature of human eggs, only a very limited number of fresh zygotes were available for scientific research from a single patient. This scarcity indeed posed a challenge in conducting comprehensive experimental investigations. As documented in the published data (Zhang *et al*, 2021), without the application of pre-pronuclear transfer (PPNT) technology, obtaining embryos for transplantation was not possible (Response Table 1). Despite the constraint on the number of embryos available for the PPNT test, we were able to obtain a high-quality embryo (5AB) for potential future transplantation (Table S1, Now in Appendix Table S1), which was subsequently utilized for the construction of ESCs.

We appreciate your concern regarding the statement in the abstract and have revised it for accuracy (Page 2, Line 34-38). Additionally, we have addressed the need for additional trials through collaboration with reproductive centers to further assess the benefits of PPNT (Page 18, Line 378-381). Your thorough review of our work is highly valued, and we hope that these amendments effectively address your concerns.

Mutation	Age (Years)	Duration of infertility (Years)	IVF/ICSI cycles	Retrieved oocytes	Mature oocyte	Fertilized oocytes	Fertilized oocytes states on Day 1	Embryos that could be transferred
R379Q	28	3.5	1- IVF	11	10	10	8 in PN, 2 in 1C	0
			2-ICSI	15	13	9	1 in PN, 8 in 1C	0
			3-ICSI	8	5	5	5 in PN	0
F441fs*16	31	7	1-IVF	13	12	11	11 in PN	0
			2-ICSI	12	9	8	6 in PN, 2 in 1C	0
			3-ICSI	7	7	6	6 in PN	0

* IVF, in vitro fertilization; ICSI, intracytoplasmic sperm injection; Day 1, the first cleavage day; PN, pronucleus; C, cell.

Response Table 1. Oocyte and embryo characteristics of IVF and ICSI cycles in the patients with CHK1 mutations.

3. (lines 149-157):

- The difference in F-actin quantity is not very clear in the pictures (Fig. 2C,D). I suggest moving Suppl. Fig. 2C to the main text to underline the difference around the pronuclei rather than in the cytoplasm. Additionally, the Methods section lacks information on methods of quantification.

- Authors claim that mutant CHK1 zygotes can be characterized by an increased fraction of CHK1 in the cytoplasm and they support this by citing Fig. 2C and 2E whereas these panels do not show this.

Re:

Thank you for your insightful feedback on our manuscript, specifically regarding the clarity of F-actin quantity and CHK1 fraction in the images and the need for additional information on quantification methods. We appreciate your suggestions and have made the following modifications to address your concerns:

To enhance the clarity of the difference in F-actin quantity, we effectively illustrated the distribution of F-actin signal through a comprehensive line profile that extends across pronuclei of zygotes, as visually depicted in Response Figure 2 (Figure S2C,

Now in Figure EV2). And following your valuable suggestion, we have moved Suppl. Fig. 2C to the main text which we can see in Expanded View Figure 2C, to underline the difference around the pronuclei rather than in the cytoplasm. Additionally, we have revised the Methods section to include detailed information on the methods of quantification for F-actin and CHK1 signal (Page 27, Line 585-592).

We acknowledge your attention to the ambiguity surrounding the assertion of an augmented fraction of CHK1 in the cytoplasm. Panels Fig. 2C and 2E do not distinctly illustrate this, possibly due to the significantly diminished EGFP-CHK1 signal in the p.F441fs*16 group. The line profile extending across zygotes in the Response Figure 2 also offers a more precise and detailed depiction of the observed disparities in CHK1 localization. To enhance transparency and clarity regarding the quantitative aspects of our analysis, we have provided additional explanations in the manuscript (Page 7, Line 156-161). We value your constructive feedback and hope that these revisions address your concerns adequately.

Response Figure 2. Line profiles across zygotes illustrating the distribution of F-actin and EGFP signal

4. (Line 212-216 and 232-236): referring to Fig. 4 and Suppl. Fig. 4F, the authors show CHK1 and MICAL3 PLA and co-localization results in GV oocytes, as well as 2-cell and 4-cell embryos. Is there any particular reason why these analyses were not completed in zygotes specifically? I would assume that it is most important to show

this stage from the point of view of this manuscript.

Re:

Thank you for your insightful observation regarding the analysis of CHK1 and MICAL3 PLA and co-localization results in the context of our manuscript. We appreciate your attention to detail and agree with the importance of including zygotes in this analysis. Due to the availability of human-specific antibodies for MICAL3, the PLA experiment could only be conducted using donated human oocytes and embryos. Unfortunately, obtaining donated human zygotes was challenging.

However, we understand your concern and acknowledge the importance of conducting PLA at the zygote stage. To address this, we utilized donated human three-pronuclear zygotes for the CHK1 and MICAL3 PLA experiment (Response Figure 3, corresponding to Appendix Figure S2; Page 11, Line 231-233 in the manuscript). Unfortunately, we encountered challenges in the colocalization experiment upon attempting to reuse the zygote after the PLA analysis. Despite this limitation, the addition of the zygote-stage embryo to our PLA analysis ensures a more comprehensive coverage of relevant developmental stages, as you rightly pointed out.

Response Figure 3. Human zygote displays significant red PLA signals.

5. (Line 202): *I would find it informative to explore more the topic of 32 proteins identified as potentially interacting with CHK1. I think that grouping them additionally by function or engagement in known processes would be very insightful.*

Re:

Thank you for your valuable suggestion regarding the exploration of the 32 proteins identified as potentially interacting with CHK1. We agree that providing additional insights into these proteins by grouping them based on function or engagement in known processes would enhance the understanding of their significance in the context of CHK1 interactions. Specifically, we have grouped them based on their functions and the MICAL3 protein is involved in the term of supramolecular fiber organization (Response Figure 4, corresponding to Figure EV4G in the manuscript). This addition aims to provide readers with a clearer understanding of the potential functional implications of CHK1 interactions with these proteins.

We appreciate your thoughtful feedback and believe that this enhancement contributes to the overall depth and comprehensiveness of our study.

Response Figure 4. Bar graph of enriched terms across proteins identified as potentially interacting with CHK1.

6. *In the introduction section, authors state that "it is common to observe embryos arresting at the two-pronuclei (2PN) stage (Rawe et al, 2003)".*

- In the cited paper the rate of the 2PN arrest is not given. However, according to Zhang et al., 2021, the rate of arrest at the zygote stage is 2%. Could the authors

provide an actual number with a reference rather than this general statement?

- How frequent is mutation of CHK1 in patients?

Re:

Thank you for your diligence in reviewing our manuscript and for pointing out the need for more specific information in the introduction section. We have addressed your concerns as follows:

We appreciate your suggestion to provide a more specific and referenced number for the rate of embryos arresting at the two-pronuclei (2PN) stage. We have revised the introduction section to include the information from Zhang et al., 2021, indicating that the rate of arrest at the zygote stage is 2%(Zamora *et al*, 2010) (Page 3, Line 61-63)

In response to your inquiry about the frequency of CHK1 mutation in patients, we have performed whole exome sequencing and Sanger sequencing analyses of 29 patients with zygote arrest, characterized by pronuclear breakdown failure. The results demonstrated that 7 patients carried CHK1 mutations in the C-terminal conserved domain, indicating a frequency of 24.1% (Zhang *et al.*, 2021).

We appreciate your feedback, and these revisions aim to improve the accuracy and specificity of the information presented in the manuscript.

7. (Line 176): Authors say that nocodazole treatment "Intriguingly" resulted in zygotes progressing to prometaphase with PNEB occurrence. This effect is not so much intriguing in my opinion as it has been known for many years both in somatic cells and embryos. Also, in Discussion section (lines 415-417), the authors claim that their study uncovered that not microtubules but actin is essential for (P)NEB which I find a little overstated, at least for involvement of microtubules.

Re:

Thank you for your thoughtful comments on the discussion surrounding the nocodazole treatment and the interpretation of the essential role of actin in PNEB. We appreciate your insights and have made the following adjustments to address your

concerns:

The term "Intriguingly" has been replaced with a more neutral expression "Subsequently" to better reflect the well-known nature of the observed effect in both somatic cells and embryos.

We have rephrased the relevant section regarding the essential role of actin in PNEB to ensure a more balanced and nuanced interpretation (Page 8, Line 180; Page 9, Line 193-196; Page 18, Line 378-381; Page 33, Line 743). We appreciate your careful review of our manuscript and hope that these changes address your concerns adequately.

8. (Line 187) There should be a reference to Fig. S3B when "Phase B" is mentioned. Otherwise, I find it confusing.

Re:

Thank you for pointing out the need for clarity regarding the mention of "Phase B", approximately 2 hours prior to occurrence of PNEB, in Fig. S3B (Now in Figure EV3B). In order to make the inhibitor administration period clearly, we have made the illustration of time line between pronuclear formation and PNEB in Fig.S3B according to the reference Scheffler et al., 2021. We appreciate your attention to detail, and we have added the reference to Fig. S3B (Now in Figure EV3B).

9. Figure 2A, 2C and other Figures: at least the polar body and chromatin should be marked in the pictures

Re:

Thank you for your suggestion to improve the clarity of our figures, specifically Figures 2A, 2C, and others, by marking the polar body and chromatin. We acknowledge the importance of providing clear visual cues for key structures in the images. To address this concern, we have revised the figures by adding clear markings especially for the pronuclei and chromatin, to enhance the visibility and understanding of these structures. However, it is important to note that in some instances, the polar bodies

(PB) were inadvertently obscured due to the angle of image collection. Consequently, we mainly focused on clearly highlighting either the pronuclei or the chromatin from pronuclei.

10. Figure 6I and 6J: quantifications have not been provided.

Re:

Thank you for bringing to our attention the missing quantifications in Figures 6I and 6J. We appreciate your thorough review, and we apologize for any oversight in the presentation of the data.

To rectify this issue, we have taken immediate action by conducting a line profile analysis across the pronuclei. This analysis specifically focuses on quantifying the signal alteration between the EGCG and control (DMSO) groups, with a particular emphasis on the region surrounding the pronuclei (Response Figure 5, We supplemented the quantification below Figure 6I and 6J). By implementing this additional quantitative approach, we aim to provide a more comprehensive and robust representation of the observed changes.

Response Figure 5. Line profiles were generated across zygotes below Figure 6I and 6J.

Referee #2:

The authors start out in intact human oocytes and determine pronuclear envelope breakdown (PNEB) as a key event indicating the start of zygote development in control/wt oocytes as compared to oocytes facing gain-of-function mutants of CHK1 kinase. Indeed, these mutants inhibit progression beyond the zygote stage. The action of the mutant is initially characterized in a remarkable pre-pronuclear transfer experiment, in which exchange of the ooplasm rescues development, suggesting that it is an early-stage cytoplasmic action of CHK1 that inhibits progression of zygote development. The authors then move one step forward and show that PNEB is likewise inhibited when compromising actin polymerization (not MT). The obvious hypothesis is then tested: is there a link between CHK1 activity and regulation of actin assembly. Identification of CHK1 interaction partners addresses this. Among several interactors, MICAL3 is picked as a candidate to explain a link between CHK1 activity and actin assembly in the zygote. Interaction studies in zygotes and somatic cells confirm the initial identification, which seems to be increased in mutant variants of CHK1. Finally, specific inhibition of MICAL3 function rescues zygote development suggesting that its inhibition by lowering CHK1 activity is required for PNEB and zygote development.

Taken together, this is a very nice piece of work and foresee publication in EMBO reports given that one central aspect will be addressed adequately. In the current version of the manuscript, the link between CHK1 action and the actin cytoskeleton. I appreciate that interaction between CHK1 and MICAL3 is documented and that inhibition of MICAL by ECGC rescues at an optimal concentration.

Re:

Thank you for your meticulous review and valuable insights into our research. We genuinely appreciate the time and effort you dedicated to evaluating our work. Your feedback has been instrumental in refining our manuscript.

However, the focus on MICLA3 is not well documented. Mass spec data come "out of the blue", i.e. there are no background control values, the number of unique peptides is low as well as the overall score. Other, more abundantly identified proteins are not further commented on. This needs to be better documented/clarified.

Re:

We sincerely apologize for any confusion regarding the presentation of the mass spectrum data. In fact, a total of 6000 mouse zygotes were collected through in-vitro fertilization and subjected to immunoprecipitation using both IgG control and CHK1 antibody. The results presented in Fig. 4A have specifically excluded proteins identified in the IgG control. To delve deeper into the mechanisms by which CHK1 regulates the F-actin meshwork in the cytoplasm of fertilized eggs, we made a concerted effort to identify a specific target of CHK1 that could serve as a crucial intermediary bridge, orchestrating the dynamics of F-actin. Among the 32 proteins likely interacting with CHK1, MICAL3 functions as a specific F-actin regulator. Consequently, we have shifted our focus to MICAL3 (Page 9-10; Line 202-205).

The limited number of unique peptides and overall score may be attributed to the rarity of zygotes compared to somatic cells. Additionally, we have provided supplementary information regarding their relevance through pathway analysis, ensuring a more comprehensive presentation of our findings (Response Figure 1, corresponding to Figure EV4G in the manuscript).

Response Figure 1. Bar graph of enriched terms across proteins identified as potentially interacting with CHK1.

Moreover, even though the outcome of the EGCG inhibition seems obvious, its specificity is neither commented nor tested at all. How do we know that what we see is specific? There should be an alternative approach with e.g. overexpression of negative (mono-oxygenase dead) mutant variants or knockdown to reduce activity of endogenous MICAL3 using an alternative to EGCG.

Re:

We deeply appreciate your thoughtful comments and invaluable suggestions. We extend our sincere apologies for any shortcomings in the precision and application of a MICAL inhibitor. In fact, we have chosen to take advantage of the morpholino oligo combining with siRNA targeting Mical3 to conduct knockdown experiments in mouse zygotes. However, the knockdown efficiency is not that obvious (Response Figure 2). We usually conduct zygotes knockdown and verify the efficiency at least at 2-cell stage, thus the low knockdown efficiency maybe due to insufficient working time.

In addition, based on the analysis of in-vitro MICAL3 monooxygenase activity, the catalytic efficiency of MICAL3 dramatically increased on adding F-actin only when the CH domain was available (Kim *et al*, 2020). We then tried the overexpression protocol by mutating two residues Glu213 and Arg530 separately in the FMO (Monooxygenases domain) and CH domains to reduce MICAL3's catalytic efficiency (Response Figure 3A)(Kim *et al.*, 2020). However, the human MICAL3 fragments show an abnormal nucleus-localization in mouse 2-cell embryos (Response Figure 3B). More researches in the future are necessary to explore the key enzyme-regulation residues of MICAL3 in an in-vivo model.

It has been reported that certain green tea polyphenols which can block flavoprotein monooxygenases, such as (-)-epigallocatechin gallate (EGCG) (Pasterkamp *et al*, 2006; Terman *et al*, 2002). The application of EGCG can significantly improve the development rate of zygote with CHK1 mutation (Figure 6A-E). Further analyzes indicate EGCG can reduce the depolymerization of F-actin around pronuclei (Figure 6I) and reduce the production of H₂O₂ (Figure 6J) in zygotes. Although disappointed

by the failed knockout/overexpression attempt, the application of EGCG to inhibit MICALs might partially underscore the role of mutant CHK1 in orchestrating H2O2 elevation and F-actin depolymerization mediated by MICAL3.

Response Figure 2. Relative mRNA expression of MICAL3 in zygotes after knockdown experiments.

Response Figure 3. Overexpression of EGFP-MICAL3 in mouse zygote with potentially decreased catalytic efficiency.

Finally, the statement of statistics are often superficial; how many independent experiments were performed and what is the actual sample size is not consistently given.

Re:

Thank you for your feedback. We have revised the manuscript to explicitly mention the number of independent experiments and provide the actual sample size for each statistical analysis in the figure legends. These updates aim to enhance the transparency and thoroughness of our statistical reporting.

Of minor importance are numerous grammatical and spelling mistakes that should and will be corrected prior to publication.

Re:

Additionally, we have conducted a thorough proofreading to address any grammatical and spelling mistakes, ensuring the manuscript meets the highest standards of language quality. Thank you once again for your valuable feedback.

Referee #3:

In the manuscript EMBOR-2023-58263V1 with the title "CHK1 Controls Zygote Pronuclear Envelope Breakdown by Regulating F-actin through Interacting with MICAL3" the authors reveal a novel function for CHK1 in controlling MICAL3 monooxygenase activity for F-actin disassembly necessary for successful Pronuclear Envelope Breakdown (PNEB) and subsequent first mitosis. In general, the authors use a variety of common techniques such as confocal microscopy, smFRET, immunoblotting, Co-IPs and in-vitro assays next to pre-pronuclear transfer (PPN) as uncommon but required technique to uncover mechanistic insights in the process of PNEB in fertilized zygotes from humans and mice as well as in the derived embryonic stem cell lines or human HEK cells. As the use of PPN has to be considered questionable in an ethical background it allows for new experimental approaches to gain insights into the developmental processes and subsequently develop future therapeutic options. The authors show remarkable development of functional blastocysts out of patients' zygotes with severe defects in embryogenesis due to CHK1 mutations prohibiting first mitotic division after fertilization. Using confocal imaging the authors show a disruption of F-actin structures around the pronuclei presents in the two-pronuclei (2PN) stage of the embryos of patients with mutant CHK1. Further experiments show actin-dependency rather than microtubule involvement mediating PNEB using specific drugs for either cytoskeletal component. Mass spectrometry analysis of CHK1 in mouse zygotes revealed the actin disassembly factor MICAL3. Proximity ligation assays in HEK cells as well as in zygotes confirmed association between CHK1 and MICAL3. Characterization of mutant CHK1 displayed an open conformation compared to wt CHK1 going hand in hand with an increased association with MICAL3 in immunoprecipitation experiments. Increased enzymatic activity of MICAL3 by mutant CHK1 leads to higher F-actin depolymerization rates. Finally, the authors partially rescue their phenotype by mutant CHK1 by inhibiting MICAL3 monooxygenase function with epigallocatechin gallate (EGCG), a MICAL monooxygenase activity inhibitor.

Overall, the authors took a great effort in putting together this data.

In summary, I support the publication of the manuscript, if the following points can be addressed by the authors:

Re:

We appreciate your thorough review of our manuscript and your positive feedback on the efforts we invested in unraveling the novel role of CHK1 in controlling MICAL3 monooxygenase activity. We would like to express our gratitude for your support and insightful comments. Your feedback has been invaluable in improving the clarity and robustness of our manuscript.

Image display in figures:

We recommend changing the displayed colors in figures in a color-blind version to avoid green and red images in the same figure. Easiest here would be to change red to magenta.

Re:

Thank you for your valuable feedback and suggestion to enhance the accessibility of our figures for color-blind readers. We appreciate your consideration of potential difficulties with red-green color perception. In response to your recommendation, we adjusted the displayed colors in the figures by changing red to magenta, especially the figures confusing in understanding with green and red composition, ensuring better differentiation for color-blind individuals.

Major points:

1) In Fig. 2 the authors say "As anticipated, zygotes carrying the CHK1 mutation (p.F441fs16) displayed a significant reduction in F-actin signal, particularly in cytoplasm and around pronuclei, compared to wild-type zygotes (Figure 2C, 2D and Figure S2C). Additionally, consistent with prior findings (Zhang et al., 2021), the expression of mutant CHK1 in the pronuclei markedly diminished, with an increased proportion exhibiting cytoplasmic localization (Figure 2C, 2E and Figure S2C)."

I do appreciate the authors quantification results, however the lower image of wt zygotes in Fig. 2C does not show an increased CHK1 localization in the two indicated pronuclei compared to mutant CHK1 and to me the cytoplasmic CHK1 wt localization compared to the image below that for mutant CHK1 do look roughly the same. To better represent their findings the authors should choose a different representative image for the lower wt panel showing CHK1 localization in the pronuclei. The linescans in Fig. S2 across the whole cell covering both of the pronuclei and plot the fluorescence intensity for CHK1-EGFP and Phalloidin provide additional useful data that could be displayed in the main figure. In general, showing the increased localization of CHK1 in the pronuclei together with an increase in F-actin fluorescence around the pronuclei and decrease in the cytoplasmic intensity in the main figure with representative images would be more suitable.

Re:

Thank you for your insightful feedback. We appreciate your suggestions to enhance the representation and clarity of our findings in Fig. 2C. We have changed a more suitable and representative image in the lower panel of wt zygotes to better illustrate the increased CHK1 localization in the pronuclei compared to mutant CHK1. Furthermore, we agree that including line profiles from Fig. S2, covering both pronuclei and plotting fluorescence intensity for CHK1-EGFP and Phalloidin, in the main figure (Now in Figure EV2C of the manuscript)

Thank you for your constructive input, and we look forward to implementing these improvements in the revised manuscript.

2) The authors performed a PLA assay in Fig. 4 for an interaction-readout between CHK1 and MICAL3, however there are some general controls missing for the cell-based PLA. As indicated in the materials and methods the authors stained without the primary antibodies as negative control in Fig. 4D. Additionally, it would be good to see PLA assays in HEK cells with the following conditions: primary antibodies individually with either secondary antibody to exclude unspecific binding of the

secondary antibody to the other primary antibody, a positive control of a known CHK1 interactor (see Blasius et al. 2011) and a knockdown of either CHK1 or MICAL3 via siRNA compared to non-targeted siRNA over three independent biological replicates to further support their findings on the interaction of CHK1 and MICAL3. In this connection, Fig. 4E shows the quantification of PLA dots per cell without cellular labeling for the cell body. It is unclear how the authors distinguished between different cells, as it looks that the quantification represents the dots of each cell in the provided representative image in Fig. 4D. There is no indication of biological replicates for this experiment as the authors wrote "Most experiments were repeated at least three times", please clarify. Please also explain the ImageJ quantification of PLA dots in the materials and methods section in more detail as there is no co-staining of a cellular marker besides DAPI for nuclear staining. Are these maximum intensity projections? if so, the authors should indicate.

Re:

Thank you for your detailed and constructive feedback. We appreciate your insightful suggestions to improve the experimental clarity of the PLA assay.

While we acknowledge the potential benefits of the proposed experiments to further support findings on the interaction of CHK1 and MICAL3, we believe that our current data sufficiently support our main findings regarding the interaction between CHK1 and MICAL3. On the one hand, the PLA experiments are strictly performed according to the protocol illustrated by the manufacturer (<https://www.sigmaaldrich.cn/CN/zh/technical-documents/protocol/protein-biology/protein-and-nucleic-acid-interactions/duolink-fluorescence-user-manual>). On the other hand, the interaction between CHK1 and MICAL3 has been further verified by the CO-IPs and protein-protein models. We hope that, despite these limitations, the current manuscript meets the standards for publication. We value your feedback and appreciate your understanding.

We apologize for any confusion in the presentation of PLA quantification. The quantification of PLA signals in HEK-293 cells relied on determining the average number of red dots per cell in each image field. To be more specific, the total PLA red dots were divided by the number of nuclei, as indicated by DAPI staining. We provided a clearer explanation in figure legends. Additionally, we clarified the indication of biological replicates, ensuring transparency in reporting experimental repetitions (Page 26, Line 556-560; Page 33-34, Line 760-762).

We also appreciate your suggestion to provide more details on the ImageJ quantification of PLA dots in the materials and methods section. What's more, the maximum intensity projections were used in confocal collection. We offered a more thorough explanation in the method section for the confocal collection method, Z-scan across the nucleus. (Page 26, Line 556-560).

Thank you for your valuable input, and we look forward to addressing these points in the revised manuscript.

3) The authors perform smFRET measurements of mutant CHK1 compared to wt CHK1 and plot FRET efficiency in Fig. 5B and C. The figure would benefit from a crop of the bleached region for the CHK1 panel for the reader to better see the fluorescence increase after successful FRET for wt CHK1. Additionally, there is not enough information about the calculations of the FRET efficiency in the materials and methods section.

Re:

Thank you for your insightful comment. In response to your suggestion, we provided a cropped region of the bleached area for the CHK1 panel in Fig. 5B (Response Figure 1) to enhance the visibility of the fluorescence changes after FRET for CHK1 signals. Moreover, we included a more detailed explanation of the FRET efficiency calculations in the Materials and Methods section to ensure transparency and clarity (see Page 26, Line 566-568; Page 34, Line 778).

Response Figure 1. Representative images demonstrate the FRET efficiency in different CHK1 groups.

4) In Fig. 5 the authors performed assays with pyrene labelled actin to quantify the depolymerization rates of CHK1 mutants compared to wt CHK1. Similar to other methods described in this paper I have some things to point out here next to the lack of information in the methods section. The authors use the commercially available kit from Cytoskeleton for their experiments but end up measuring only 1 timepoint for a kinetic assay during 1 h incubation. It would be essential to read out multiple times during their 1 h incubation time to determine exact kinetic curves for actin depolymerization in three biological replicates, as this assay is sensitive enough to provide this information (see Mu et al. 2020 PMID: 31871199). Furthermore, the author should plot their readouts similar to the paper I referred to in the last sentence and add one control to the experiment, such as the depolymerization of F-actin alone in G-Buffer compared to the addition of wt and mutant CHK1.

Re:

Thank you for your careful consideration of our manuscript and for providing constructive feedback. We have carefully reviewed your suggestion to perform additional experiments related to the depolymerization rates of CHK1 mutants compared to wild-type CHK1 using pyrene-labeled actin.

We appreciate your concern and understand the importance of obtaining kinetic curves for actin depolymerization. Actually, we have tried multiple-times signal collection during the 1 h incubation. However, the signal underwent obvious quenching and we then only selected the one-time point for analysis. We sincerely apologize for the lack of multiple-times signal collection due to practical constraints. For the clarity of the F-actin depolymerization assay, we supplemented more detailed information in the methods section (Page 27, Line 585-592).

5) The authors' aim to rescue the mutant phenotype by applying the EGCG, an inhibitor for MICAL monooxygenase activity and claim: "As anticipated, EGCG exhibited a partial restoration of the F-actin signal, particularly concentrated around the pronuclei (Figure 6I)." It is quite difficult to visually see the partial restoration of the F-actin signal around the pronuclei. A linescan analysis similar to Fig. S2 would better support their claims. In addition, applying the EGCG drug to an actin depolymerization assay showing less depolymerization of F-actin or performing the PLA assay in combination with EGCG would strengthen the authors claims and make their rescue approach more valid. If this is not possible the authors can discuss the partial rescue in more detail.

Re:

We appreciate your insightful observation regarding the visualization of the partial restoration of the F-actin signal with EGCG treatment. In response to your suggestion, we conducted a linescan analysis similar to Fig. S2 (Now in Figure EV2C), now incorporated into the revised Figure 6I for better clarity and to reinforce our claims (Response Figure 2).

Moreover, we acknowledge the importance of performing the F-actin rescue experiment in the zygote model rather than HEK-293 cells. However, conducting the actin depolymerization assay in zygotes would require a substantial quantity, posing challenges in collection. Additionally, due to the unavailability of mouse-specific antibodies for MICAL3, we were unable to perform the PLA assay in combination with EGCG.

In an alternative approach, the application of EGCG demonstrates a significant enhancement in the developmental rate of zygotes with CHK1 mutation, as illustrated in Figure 6A-E. To substantiate the association between the rescue effect and the modified activity of MICAL3, we conducted an analysis of F-actin distribution and H₂O₂ signals in zygotes following EGCG administration. Subsequent analyses indicate that EGCG effectively mitigates the depolymerization of F-actin around pronuclei (Figure 6I) and reduces the production of H₂O₂ (Figure 6J) in zygotes. It is noteworthy that the observed alteration in F-actin distribution may not be as pronounced, potentially attributed to the specific time frame of EGCG administration or variations in fluorescence efficiency.

Response Figure 2. Line profiles were generated across zygotes in Figure 6I.

Minor points:

6) In Fig. 4 B and C the authors should indicate the molecular weight. Furthermore, if they can the authors should add a loading control to Western blot analysis, as the amount of pulled down protein levels are fairly low compared to the input thereby

displaying a weak interaction/association under basal and closed conformation conditions. How do the authors explain the significant PLA dot increase of CHK1 and MICAL3 in their cell-based PLA assay compared to the weak interaction in their Co-IPs under basal conditions?

Re:

Thank you for your thoughtful comments. We acknowledge the importance of indicating molecular weights in Fig. 4 B and C. In the revised version of the manuscript, these details have been included for clarity.

We apologize for the absence of a loading control, and after careful consideration, we acknowledge that suitable loading controls are challenging to identify in this analysis. In addition, it's essential to note that PLA signals typically manifest as distinct dots, resulting from the amplification of PLA probes ligated to antibodies targeting a pair of interacting proteins. We assume that the interaction between wild-type CHK1 and MICAL3 is relatively weaker. Given the signal amplification process of the PLA assay, it shows a more obvious interaction between CHK1 and MICAL3 than the CO-IP method.

7) Fig. 5D is missing the molecular weight indication and a loading control for CHK1 IP blot (right panel) is essential here if the authors want to claim an "enhancement" of mutant CHK1 and MICAL3 interaction/association compared to wt CHK1. If the GAPDH loading control of the input panel also refers to the CHK1 IP panel the authors can neglect this point and provide an image of the whole membrane in their supplementary information for transparency reasons.

Re:

Thank you for your valuable feedback. We appreciate your attention to detail. In the revised manuscript, we ensured the inclusion of molecular weight indications for Fig. 5D. Besides, the GAPDH loading control of the input panel also corresponds to the CHK1 IP panel. Additionally, we have presented the entire membrane image for transparency according to the journal's policy.

Reference

Beaudouin J, Gerlich D, Daigle N, Eils R, Ellenberg J (2002) Nuclear envelope breakdown proceeds by microtubule-induced tearing of the lamina. *Cell* 108: 83-96

Hong J, Hu K, Yuan Y, Sang Y, Bu Q, Chen G, Yang L, Li B, Huang P, Chen D *et al* (2012)

CHK1 targets spleen tyrosine kinase (L) for proteolysis in hepatocellular carcinoma. *J Clin Invest* 122: 2165-2175

Kim J, Lee H, Roh YJ, Kim HU, Shin D, Kim S, Son J, Lee A, Kim M, Park J *et al* (2020)

Structural and kinetic insights into flavin-containing monooxygenase and calponin-homology domains in human MICAL3. *IUCrJ* 7: 90-99

López-Contreras AJ, Gutierrez-Martinez P, Specks J, Rodrigo-Perez S, Fernandez-Capetillo

O (2012) An extra allele of Chk1 limits oncogene-induced replicative stress and promotes transformation. *The Journal of experimental medicine* 209: 455-461

Mühlhäusser P, Kutay U (2007) An in vitro nuclear disassembly system reveals a role for the

RanGTPase system and microtubule-dependent steps in nuclear envelope breakdown. *J Cell Biol* 178: 595-610

Pasterkamp RJ, Dai HN, Terman JR, Wahlin KJ, Kim B, Bregman BS, Popovich PG, Kolodkin

AL (2006) MICAL flavoprotein monooxygenases: expression during neural development and following spinal cord injuries in the rat. *Molecular and cellular neurosciences* 31: 52-69

Sriuranpong V, Mutirangura A, Gillespie JW, Patel V, Amornphimoltham P, Molinolo AA,

Kerekhanjanarong V, Supanakorn S, Supiyaphun P, Rangdaeng S *et al* (2004) Global gene expression profile of nasopharyngeal carcinoma by laser capture microdissection and complementary DNA microarrays. *Clin Cancer Res* 10: 4944-4958

Terman JR, Mao T, Pasterkamp RJ, Yu HH, Kolodkin AL (2002) MICALs, a family of

conserved flavoprotein oxidoreductases, function in plexin-mediated axonal repulsion. *Cell* 109: 887-900

Verlinden L, Vanden Bempt I, Eelen G, Drijkoningen M, Verlinden I, Marchal K, De Wolf-Peeters C, Christiaens MR, Michiels L, Bouillon R *et al* (2007) The E2F-regulated gene Chk1 is highly expressed in triple-negative estrogen receptor /progesterone receptor /HER-2 breast carcinomas. *Cancer Res* 67: 6574-6581

Xu J, Li Y, Wang F, Wang X, Cheng B, Ye F, Xie X, Zhou C, Lu W (2013) Suppressed miR-424 expression via upregulation of target gene Chk1 contributes to the progression of cervical cancer. *Oncogene* 32: 976-987

Zamora RB, Sánchez RV, Pérez JG, Díaz RR, Quintana DB, Bethencourt JCA (2010) Human zygote morphological indicators of higher rate of arrest at the first cleavage stage. *Zygote* 19: 339-344

Zhang H, Chen T, Wu K, Hou Z, Zhao S, Zhang C, Gao Y, Gao M, Chen ZJ, Zhao H (2021) Dominant mutations in CHK1 cause pronuclear fusion failure and zygote arrest that can be rescued by CHK1 inhibitor. *Cell Res* 31: 814-817

Sincerely,

Shigang Zhao, M.D., Ph.D.
Center for Reproductive Medicine
Shandong University
#157 Jingliu Road, Jinan 250001, China
Tel: +86-18954109360
Email: zsq0108@126.com

Dear Dr. Wu,

Thank you for submitting your revised manuscript. It has now been seen by two of the original referees, whose comments are copied below. As you can see, both referees #2 and #3 acknowledge that the manuscript improved during revision, but they also have remaining concerns. Given the interest in the findings and the support of the referees, I would like to give you another chance to revise the study to address the remaining concerns. In particular,

- The EGCG inhibition needs to be confirmed using a different small molecule inhibitor (referee #2).
- The concerns of referee #2 on the presentation of the initial mass spectrometry result need to be addressed.
- Additional controls for the PLA assays (referee #3, point 1) and actin depolymerization assays (referee #3, point 2) need to be provided.

Moreover, I need you to address the editorial points below.

- The "Competing interest statement" section needs to be renamed as "Disclosure Statement and Competing Interests" and placed after Acknowledgments.
- As per EMBO Press policy, please provide institutional email addresses of the dual corresponding authors Dr. Wu and Dr. Zhao.
- Please remove the "Author contributions" section from the manuscript.
- Please fill out and include an author checklist as listed in our online guidelines (<https://www.embopress.org/page/journal/14693178/authorguide>)
- The funding information is currently incomplete in the manuscript submission system - i.e. the following is missing: 82071606, 82171842; Basic Science Center Program of NSFC, Grant/Award Number: 31988101; CAMS Innovation Fund for Medical Sciences, Grant/Award Number: 2021-I2M-5-001; Shandong Provincial Key Research and Development Program, Grant/Award Number: 2020ZLYS02; Taishan Scholars Program of Shandong Province, Grant/Award Number: ts20190988; the National Key Research and Development Program of China, Grant/Award Number: 2021YFC2700400; Postdoctoral Research Fund of Gusu School in Nanjing Medical University, Grant/Award Number: GSBSHKY202303; Innovative research team of high-level local universities in Shanghai, Grant/Award Number: SHSMU-ZLCX20210200, Fundamental Research Funds of Shandong University (2023QNTDO04). Please note that you can use "More Funders" button if you run out of the designated entries.
- Please submit the Appendix file in pdf format. Please remove the last page (empty).
- We note the source data folder of Figure 5B is empty.
- During our routine figure checks, we note a potential re-use between Figure 2C F-actin and Figure EV2 F-actin and a potential cell re-use between Figure 4G DAPI and Appendix Figure S1 DAPI/merge, which is only allowed if the images/cells are derived from the same experiment, in which case it should be clarified in the legends of all respective figure panels.
- The study employs human oocytes. For experiments involving human subjects the corresponding author must identify the committee approving the experiments and include a statement that informed consent was obtained from all subjects and that the experiments conformed to the principles set out in the WMA Declaration of Helsinki and the Department of Health and Human Services Belmont Report. (<https://www.embopress.org/page/journal/14693178/authorguide#humansubjects>)
- The Data Availability section is reserved for the primary datasets generated in the study. If your study does not include datasets, please insert the following statement: This study includes no data deposited in external repositories.
- Our production/data editors have asked you to clarify several points in the figure legends:
 - o Please note that a separate 'Data Information' section is required in the legends of figures 2d-e; 5e-f; 6i-j; EV 4d-f.
 - o Please note that the legends for figures EV 1f-g are missing in the manuscript. This needs to be rectified.
 - o Please note that information related to n is missing in the legends of figures 6d-e, g-h; EV 1d; EV 4a-c.
 - o Please note that the error bars are not defined in the legends of figures EV 1d; EV 4a-c.
 - o Please note that the scale bar needs to be defined for figures EV 2c.
 - o Please note that scale bar and its definition are missing for figure 5b.
 - o Please note that the white dashed circles are not defined in the legend of figure 5g. This needs to be rectified.
- Papers published in EMBO Reports include a 'synopsis' and 'bullet points' to further enhance discoverability. Both are displayed on the html version of the paper and are freely accessible to all readers. The synopsis includes a short standfirst summarizing the study in 1 or 2 sentences (max 35 words) that summarize the paper and are provided by the authors and streamlined by the handling editor. I would therefore ask you to include your synopsis blurb and 3-5 bullet points listing the key experimental findings.
- The synopsis image needs to be 550px wide and 300-600px high. When your synopsis image is resized accordingly, some of the labels are too small to read (please see attached). Please provide a synopsis image with larger labels.

Thank you again for giving us to consider your manuscript for EMBO Reports, I look forward to your revision.

Kind regards,

Deniz Senyilmaz Tiebe

--
Deniz Senyilmaz Tiebe, PhD
Editor
EMBO Reports

Referee #2:

Zhang et al. hand in their manuscript "CHK1 controls controls Zygote Pronuclear EB by regulating F-actin through interacting with MICAL3" in revised version.

Being generally supportive for publication due to the overall significance of the study, I raised two issues that should have been addressed prior to publication.

I understand that further, independent evidence for the inhibition of MICAL (e.g. via knockdown or overexpression of dominant negative variants) beyond inhibition using EGCG turned out to be not suitable or insufficient to significantly reduce protein amounts. I do understand these efforts did not yield further evidence for the authors' idea of MICAL function to regulate the oocytes' actin cytoskeleton. In turn, I think it would be / have been even more important to confirm EGCG inhibition using a second small molecule inhibitor.

The second issue concerns the presentation of the protein identification data from the initial mass spectrometry result. I fully appreciate that this has been a tedious experiment (6,000 oocytes collected manually) and therefore could not be evaluated in a second replicate. To my opinion, however, it is even more important to display and explain the data in a transparent manner. Fig. 4A still just shows us a list of apparently identified proteins with non-defined "protein scores" and based on x peptides identified. The list ends with 32 proteins apparently cut by using a certain threshold. The authors have subtracted IgG controls, but no further details are given in the method section, figure legend or results part. Which peptides were identified, what are probability values, sequence coverage, signals above noise or absence in IgG controls, respectively. Why not show the original identification data at least for MICAL, compared to the no-show in the IgG control? Fig. 4B nicely evaluates the interaction using reciprocal IP and it would be a pity if the protein ID did not give us solid data at least on the identification of MICAL3. Frankly, the way it is presented now, raises more doubts than generates convincing evidence.

I am still very much in favor of publication, but homework needs to be done properly prior to that.

Referee #3:

The authors have addressed some of my comments by making the proposed adjustments in their display items such as applying changes for colorblind readers as well as showing a more suitable representative image and zoom-in areas as well as providing more information in their Materials and Methods section for their quantifications.

Unfortunately however, there is no experimental data regarding my comments on the PLA assay controls nor the pyrene-labelled actin depolymerization assay. This would be of particular importance to show an improvement of the initially submitted manuscript regarding overall specificity of MICAL3 interaction as well as the following monooxygenase activity of MICAL3 on F-actin depolymerization.

1. I agree with the authors that there is good evidence shown by the Co-IP and proteomics data to claim an interaction between MICAL3 and CHK1. However, I asked for crucial internal controls for the PLA assay to show specificity of their desired antibodies and functionality of their assay, which is essential for this experimental approach and has not been shown for either the cell-based (HEK cells) PLA nor the GV, 2-cell and 4-cell stadium. The authors refer to their experiment as strictly performed according to the manufacturers protocol. However, this protocol incorporates a Link on "how to optimize the Duolink Proximity Ligation assay" from the same company. The manufacturer even lists technical as well as biological controls that should be included in every Duolink PLA assay, similar to the ones I requested. Primary antibodies only control, as well as a positive control in HEK cells should be the minimum the authors have to do for this experiment if they want to include the PLA data in their manuscript.

2. While I do appreciate the authors explanation for quenching of the signal, I question the selection of endpoint measurements for this kinetic assay. Why was the 1 h timepoint chosen, why was the signal quenching gone by then? To me this sounds more like an optimization problem of the assay itself and this questions the actual signal the authors read out after 1 h incubation. I do not see a problem why this should not work technically as there is as an optimization guide for troubleshooting dealing with signal collection problems. It would be essential to also show actin depolymerization alone and compare the addition of wt and mutant CHK1 to actin depolymerization alone or implement a positive control for actin depolymerization as I proposed already as F-actin alone depolymerizes to a great extent after 35 min (see below Fig. 2d).

The overall experimental evidence on MICAL3 activity on actin depolymerization which is necessary for subsequent PNEB in this assay is in my opinion not sufficient. I suggest to incorporate the requested controls or to remove the panel Fig. 5 F .

Overall, I think the story is still very interesting and the authors have put some effort in this revision. However, the manuscript needs some controls in the above-mentioned experiments to further strengthen the authors claims on a specific CHK1-MICAL3 interaction and subsequent MICAL3-dependent F-actin depolymerization for PNEB.

RE: EMBOR-2023-58263V2

July 8, 2024

Dr. Deniz Senyilmaz Tiebe

Scientific Editor

EMBO Reports

Dear Editor and Reviewers,

We extend our sincere gratitude for the invaluable comments and constructive suggestions provided for our manuscript titled "CHK1 Controls Zygote Pronuclear Envelope Breakdown by Regulating F-actin through Interacting with MICAL3 (EMBOR-2023-58263V2)." Additionally, we deeply appreciate the detailed modification strategies that have proven immensely beneficial in enhancing the manuscript's quality.

We have diligently addressed each of the points raised by the editor and reviewers, and we present a summary of our responses below. Comments from the editor and reviewers are in blue, while responses from us (authors) are in black. Moreover, the changes made in the main manuscript are marked in yellow. We genuinely hope that our efforts adequately meet your expectations and requirements.

Editor's and Reviewers' comments:

Editor:

Thank you for submitting your revised manuscript. It has now been seen by two of the original referees, whose comments are copied below. As you can see, both referees #2 and #3 acknowledge that the manuscript improved during revision, but they also have remaining concerns. Given the interest in the findings and the support of the referees, I would like to give you another chance to revise the study to address the remaining concerns. In particular,

- The EGCG inhibition needs to be confirmed using a different small molecule inhibitor

(referee #2).

- The concerns of referee #2 on the presentation of the initial mass spectrometry result need to be addressed.

- Additional controls for the PLA assays (referee #3, point 1) and actin depolymerization assays (referee #3, point 2) need to be provided.

Re:

Thank you for providing the feedback from referees #2 and #3 regarding the revised manuscript. We appreciate the opportunity to address their remaining concerns and further improve the quality of our study.

Regarding the first concern raised by referee #2 on the need to confirm EGCG inhibition using a different small molecule inhibitor, we have applied an alternative small molecule inhibitor, CCG-1423. Additionally, we have carefully addressed the concerns raised by referee #2 regarding the presentation of the initial mass spectrometry result. Furthermore, we acknowledge the suggestions from referee #3 regarding the need for additional controls in the PLA assays and actin depolymerization assays and have made some improvements. Please see the details provided in the following revision responses.

Moreover, I need you to address the editorial points below.

- The "Competing interest statement" section needs to be renamed as "Disclosure Statement and Competing Interests" and placed after Acknowledgments.

Re:

We have renamed the "Competing interest statement" section to "Disclosure Statement and Competing Interests" and place it after the Acknowledgments section as requested. Thank you for the clarification.

- As per EMBO Press policy, please provide institutional email addresses of the dual corresponding authors Dr. Wu and Dr. Zhao.

Re:

Thank you for bringing this to our attention. In accordance with EMBO Press policy, we provide institutional email addresses for the dual corresponding authors, Dr. Wu and Dr. Zhao in the title page.

- Please remove the "Author contributions" section from the manuscript.

Re:

The "Author contributions" section was removed from the manuscript. Thanks for the instruction.

- Please fill out and include an author checklist as listed in our online guidelines (<https://www.embopress.org/page/journal/14693178/authorguide>)

Re:

Thank you for reminding us. We include the author checklist as outlined in your online guidelines. This helps ensure that all necessary components are properly addressed and included in the revised manuscript.

- The funding information is currently incomplete in the manuscript submission system i.e. the following is missing: 82071606, 82171842; Basic Science Center Program of NSFC, Grant/Award Number: 31988101; CAMS Innovation Fund for Medical Sciences, Grant/Award Number: 2021-I2M-5-001; Shandong Provincial Key Research and Development Program, Grant/Award Number: 2020ZLYS02; Taishan Scholars Program of Shandong Province, Grant/Award Number: ts20190988; the National Key Research and Development Program of China, Grant/Award Number: 2021YFC2700400; Postdoctoral Research Fund of Gusu School in Nanjing Medical University, Grant/Award Number: GSBSHKY202303; Innovative research team of

high-level local universities in Shanghai, Grant/Award Number: SHSMU-ZLCX20210200, Fundamental Research Funds of Shandong University (2023QNTD004). Please note that you can use "More Funders" button if you run out of the designated entries.

Re:

Thank you for providing the missing funding information. We will ensure that all the mentioned grant/award numbers are included in the manuscript submission system. If necessary, we will utilize the "More Funders" button to accommodate additional entries. We appreciate your attention to detail and apologize for any oversight in this matter.

- Please submit the Appendix file in pdf format. Please remove the last page (empty).

Re:

We will submit the Appendix file in PDF format as requested. Additionally, we have removed the last page, ensuring it is empty before submission. Thank you for your guidance.

- We note the source data folder of Figure 5B is empty.

Re:

Thank you for bringing this to our attention. We ensured that the source data folder for Figure 5B is properly populated with the necessary data. If any issues persist, we will promptly address them and provide the required data.

- During our routine figure checks, we note a potential re-use between Figure 2C F-actin and Figure EV2 F-actin and a potential cell re-use between Figure 4G DAPI and Appendix Figure S1 DAPI/merge, which is only allowed if the images/cells are

derived from the same experiment, in which case it should be clarified in the legends of all respective figure panels.

Re:

Thank you for bringing this to our attention. We carefully reviewed the images in Figure 2C, Figure EV2, Figure 4G, and Appendix Figure S1 to ensure compliance with the guidelines regarding image and cell reuse. Images in Figure 2C and Figure EV2, are indeed derived from the same experiment, we thus clarified this in the legends of Figure EV2. While Figure 4G and Appendix Figure S1 are derived from different experiment repeats, we sincerely apologize for the misunderstanding and has changed the control in Appendix Figure S1. We appreciate your thoroughness in ensuring the integrity of the manuscript.

- The study employs human oocytes. For experiments involving human subjects the corresponding author must identify the committee approving the experiments and include a statement that informed consent was obtained from all subjects and that the experiments conformed to the principles set out in the WMA Declaration of Helsinki and the Department of Health and Human Services Belmont Report. (<https://www.embopress.org/page/journal/14693178/authorguide#humansubjects>)

Re:

Thank you for highlighting this requirement. We ensure that the corresponding author identifies the committee approving the experiments involving human oocytes. We also include a statement in the method section, confirming that informed consent was obtained from all subjects and that the experiments conformed to the principles set out in the WMA Declaration of Helsinki and the Department of Health and Human Services Belmont Report. Thank you for your attention to detail in ensuring compliance with ethical standards.

- The Data Availability section is reserved for the primary datasets generated in the

*study. If your study does not include datasets, please insert the following statement:
This study includes no data deposited in external repositories.*

Re:

Thank you for clarifying the requirement regarding the Data Availability section. Given our study does not include datasets, we thus insert the statement: "This study includes no data deposited in external repositories." in the Data availability section. Thank you for your guidance on this matter.

- Our production/data editors have asked you to clarify several points in the figure legends:

o Please note that a separate 'Data Information' section is required in the legends of figures 2d-e; 5e-f; 6i-j; EV 4d-f.

o Please note that the legends for figures EV 1f-g are missing in the manuscript. This needs to be rectified.

o Please note that information related to n is missing in the legends of figures 6d-e, g-h; EV 1d; EV 4a-c.

o Please note that the error bars are not defined in the legends of figures EV 1d; EV 4a-c.

o Please note that the scale bar needs to be defined for figures EV 2c.

o Please note that scale bar and its definition are missing for figure 5b.

o Please note that the white dashed circles are not defined in the legend of figure 5g. This needs to be rectified.

Re:

Thank you for providing these detailed points from the production/data editors. We have ensured that all necessary clarifications and additions are made to the figure legends as follows:

o For figures 2d-e, 5e-f, 6i-j, and EV 4d-f, we included a separate 'Data Information' section in the legends as required.

- o We rectified the missing legends for figures EV 1f-g to the manuscript.
- o Information related to 'n' was included in the legends of figures 6d-e, g-h; EV 1d; EV 4a-c.
- o Error bars have been defined in the legends of figures EV 1d; EV 4a-c.
- o The scale bar was defined for figure EV 2c and for figure 5b.
- o The white dashed circles in the legend of figure 5g were defined accordingly.

We appreciate the attention to detail from the production/data editors and sincerely thanks for bringing these points to our attention, please don't hesitate to contact us if any information is required.

- Papers published in EMBO Reports include a 'synopsis' and 'bullet points' to further enhance discoverability. Both are displayed on the html version of the paper and are freely accessible to all readers. The synopsis includes a short standfirst summarizing the study in 1 or 2 sentences (max 35 words) that summarize the paper and are provided by the authors and streamlined by the handling editor. I would therefore ask you to include your synopsis blurb and 3-5 bullet points listing the key experimental findings.

Re:

Thank you for providing this information regarding the inclusion of a synopsis and bullet points in EMBO Reports publications. We have included a synopsis blurb summarizing the study, as well as the bullet points listing the key experimental findings before the introduction section. We appreciate your guidance in enhancing the discoverability of our paper and incorporated these elements into the revised manuscript accordingly.

- The synopsis image needs to be 550px wide and 300-600px high. When your synopsis image is resized accordingly, some of the labels are too small to read

(please see attached). Please provide a synopsis image with larger labels.

Re:

Thank you for the information regarding the size requirements for the synopsis image and the feedback regarding readability. We appreciate your attention to detail and have provided a synopsis image with larger labels to ensure readability.

Thank you again for giving us to consider your manuscript for EMBO Reports, I look forward to your revision.

Re:

We appreciate the opportunity to revise our manuscript for EMBO Reports again. We have worked diligently to address all the concerns and suggestions raised by the reviewers and editorial team. Please feel free to reach out if you have any further questions or require clarification on any aspect of the revisions.

Referee #2:

Zhang et al. hand in their manuscript "CHK1 controls Zygote Pronuclear EB by regulating F-actin through interacting with MICAL3" in revised version.

Being generally supportive for publication due to the overall significance of the study, I raised two issues that should have been addressed prior to publication.

I understand that further, independent evidence for the inhibition of MICAL (e.g. via knockdown or overexpression of dominant negative variants) beyond inhibition using EGCG turned out to be not suitable or insufficient to significantly reduce protein amounts. I do understand these efforts did not yield further evidence for the authors' idea of MICAL function to regulate the oocytes' actin cytoskeleton. In turn, I think it would be / have been even more important to confirm EGCG inhibition using a second small molecule inhibitor.

Re:

Thank you for your thoughtful review and for recognizing the overall significance of our study. We appreciate your feedback and work hard to address the important issue regarding confirming the EGCG inhibition using a second small molecule.

We firstly searched for other feasible small molecule inhibitors. However, we found that EGCG was the main inhibitor applied to inhibit the monooxygenase activity of MICALs[1-4]. Besides EGCG, we further obtained another two inhibitors reported to be associated with MICAL1 or MICAL2 activity, respectively. One is Levosimendan, a cardiac calcium sensitizer, which has been applied to inhibit the actin oxidizing activity of MICAL-1 in the presence of 100 μ M[5]. The other one is CCG-1423, a small molecule inhibitor of SRF/MRTF-A-dependent transcription[6]. CCG-1423 binds MICAL-2 and inhibits its enzymatic activity by 75% with 5 μ M treatment[6].

Considering the similar monooxygenase domain among MICALs (Response Figure 1A), we then test their effect on the zygote development. Firstly, we treated the zygotes over-expressed with wild-type CHK1 with Levosimendan or CCG-1423 separately under different concentration, to exclude the toxic influence of the inhibitors.

However, the Levosimendan treatment cause zygote arrest when the embryos in DMSO group develop into 2-cell embryos (Response Figure 1B). As for CCG-1423, its treatment doesn't significantly disturb the embryo development, with the exception of 100 μ M group (Response Figure 1B). We thus applied CCG-1423 as the candidate inhibitor to verify the inhibition of EGCG. After CCG-1423 treatment, especially under 0.1 μ M treatment, most of the zygotes carrying the CHK1 mutation (p.F441fs*16) could successfully develop into blastocyst stage (Response Figure 1C, we added it to Appendix Figure S5C-D).

The N-terminal domain of MICAL3 bears a closer resemblance to that of MICAL2 rather than MICAL1 (Response Figure 1A), suggesting that CCG-1423 could potentially exert its function on both MICAL2 and MICAL3. We have performed the molecular docking between the N-terminal domain of MICAL3 (PDB:6ICI) by the Autodock software, and find a binding pocket between the two molecular (Binding Energy: -9.3 Kcal/mol) (Response Figure 2B, we added it to Appendix Figure S5F). In addition, the docking result also indicate a binding pocket between MICAL3 and EGCG (Binding Energy: -7.43 Kcal/mol) (Response Figure 1A, we added it to Appendix Figure S5E). All in all, the EGCG inhibition effect could be replicated by CCG-1423, a small inhibitor can inhibit the monooxygenase activity of the N-terminal domain of MICAL2 and potentially of MICAL3 due to the domain similarity.

We believe that providing supplementary evidence through the use of a second inhibitor would strengthen the validity of our findings and enhance the robustness of our conclusions. Thanks again for your valuable feedback and for your support of our manuscript. Please let us know if there are any other aspects of the manuscript that require clarification.

Reference

- [1] Prifti E, Tsakiri EN, Vourkou E, *et al.* Mical modulates tau toxicity via cysteine oxidation in vivo. *Acta neuropathologica communications*, 2022, 10: 44
- [2] Nadella M, Bianchet MA, Gabelli SB, *et al.* Structure and activity of the axon guidance protein mical. *Proc Natl Acad Sci U S A*, 2005, 102: 16830-16835

- [3] Pasterkamp RJ, Dai HN, Terman JR, *et al.* Mical flavoprotein monooxygenases: Expression during neural development and following spinal cord injuries in the rat. *Molecular and cellular neurosciences*, 2006, 31: 52-69
- [4] Terman JR, Mao T, Pasterkamp RJ, *et al.* Micals, a family of conserved flavoprotein oxidoreductases, function in plexin-mediated axonal repulsion. *Cell*, 2002, 109: 887-900
- [5] Mannherz HG, Budde H, Jarkas M, *et al.* Reorganization of the actin cytoskeleton during the formation of neutrophil extracellular traps (nets). *European journal of cell biology*, 2024, 103: 151407
- [6] Lundquist MR, Storaska AJ, Liu TC, *et al.* Redox modification of nuclear actin by mical-2 regulates srf signaling. *Cell*, 2014, 156: 563-576

Response Figure 1. Verify EGCG inhibition by another inhibitor, CCG-1423. (A)

The demonstration of main domains among MICALs. (B) Testing the toxic effects of Levosimendan and CCG-1423 on the development of zygotes carrying wildtype CHK1. (C) CCG-1423 could replicate the effect of EGCG. Scale bar: 100 μm.

Two-tailed Student's t-tests. ns, no significant difference. **P < 0.01. Error bar, SEM.

Response Figure 2. Demonstration of the binding pocket between MICAL3 and EGCG (A) or CCG-1423 (B). The yellow dashed lines indicate the polar contacts between the inhibitor and the binding pocket of MICAL3 (PDB:6IC1).

The second issue concerns the presentation of the protein identification data from the initial mass spectrometry result. I fully appreciate that this has been a tedious experiment (6,000 oocytes collected manually) and therefore could not be evaluated in a second replicate. To my opinion, however, it is even more important to display and explain the data in a transparent manner. Fig. 4A still just shows us a list of apparently identified proteins with non-defined "protein scores" and based on x peptides identified. The list ends with 32 proteins apparently cut by using a certain threshold. The authors have subtracted IgG controls, but no further details are given in the method section, figure legend or results part. Which peptides were identified,

what are probability values, sequence coverage, signals above noise or absence in IgG controls, respectively. Why not show the original identification data at least for MICAL, compared to the no-show in the IgG control? Fig. 4B nicely evaluates the interaction using reciprocal IP and it would be a pity if the protein ID did not give us solid data at least on the identification of MICAL3. Frankly, the way it is presented now, raises more doubts than generates convincing evidence.

Re:

Thank you for your detailed feedback regarding the presentation of the protein identification data from the initial mass spectrometry results. We understand your concerns and agree that transparency and clarity in presenting experimental data are essential for establishing the validity of our findings.

We apologize for any confusion caused by the lack of detailed information on protein identification criteria and data interpretation. On the one hand, we have supplemented the source data to clearly display and explain the data in a transparent manner, as we can see from the Response Table 1. On the other hand, we have revised the figure legend section to provide a more thorough explanation of the protein identification process and original data source.

Thank you again for your valuable feedback, which will help improve the transparency and rigor of our study. Please let us know if there are any other aspects of the manuscript that require clarification.

Accession	Protein names	Gene names	MW [kDa]	Protein score	Sequence coverage (%)	Unique Peptides	# Peptides	# PSMs
P10761	Zona pellucida sperm-binding protein 3	Zp3	46.27	136.23	14.39	8	8	14
P68372	Tubulin beta-4B chain	Tubb4b	49.80	89.55	14.83	5	5	5
Q6NXH9	Keratin, type II cytoskeletal 73	Krt73	58.87	82.08	4.27	2	2	2
Q9QWL7	Keratin, type I cytoskeletal 17	Krt17	48.13	77.64	12.47	5	6	6
P61979	Heterogeneous nuclear ribonucleoprotein K	Hnmpk	50.94	74.69	2.59	1	1	1
Q4VAA2	Protein CDV3	Cdv3	29.71	46.99	6.41	1	1	1
Q80TB7	Zinc finger SWIM domain-containing protein 6	Zswim6	133.02	45.58	0.83	1	1	1
Q80W22	Threonine synthase-like 2	Thsnl2	54.15	42.34	1.24	1	1	1
P57776	Elongation factor 1-delta	Eef1d	31.27	42.24	17.44	3	3	3
Q9JHX2	Transcription factor Sp5	Sp5	42.03	40.93	1.76	1	1	1
Q64GA5	Cytosolic phospholipase A2 gamma	Pla2g4c	67.84	39.83	2.18	1	1	1
P02301	Histone H3.3C	H3-5	15.31	38.68	5.15	1	1	1
Q05920	Pyruvate carboxylase, mitochondrial	Pc	129.60	38.02	0.85	1	1	1
O88491	Histone-lysine N-methyltransferase, H3 lysine-36 specific	Nsd1	283.91	38.00	0.27	1	1	1
Q9D820	Prolyl-tRNA synthetase associated domain-containing protein 1	Prorsd1	18.85	37.67	4.14	1	1	1
E9Q4S1	High affinity cAMP-specific and IBMX-insensitive 3',5'-cyclic phosphodiesterase 8B	Pde8b	96.68	37.00	0.69	1	1	1
Q9ET26	E3 ubiquitin-protein ligase RNF114	Rnf114	25.73	36.41	2.62	1	1	1
P28028	Serine/threonine-protein kinase B-raf	Braf	88.72	36.26	0.75	1	1	1
Q8COW1	Ankyrin repeat and MYND domain-containing protein 1	Ankmy1	101.93	35.97	0.66	1	1	1
Q8CL19	[F-actin]-monooxygenase MICAL3	Mical3	223.58	35.28	0.35	1	1	1
Q9ESE1	Lipopolysaccharide-responsive and beige-like anchor protein	Lrba	316.86	35.23	0.46	1	1	1
A2AAJ9	Obscurin	Obscn	965.99	34.58	0.07	1	1	1
E9PVA8	eIF-2-alpha kinase activator GCN1	Gcn1	292.83	34.37	0.26	1	1	1
Q3URY6	Armadillo repeat-containing protein 2	Armc2	95.25	33.86	0.70	1	1	1
Q91YR5	cEF1A lysine and N-terminal methyltransferase	EEF1AKNMT	78.71	33.81	1.00	1	1	1
Q8VCF1	Soluble calcium-activated nucleotidase 1	Cant1	45.62	32.82	2.98	1	1	1
Q68FD5	Clathrin heavy chain 1	Ctfc	191.43	32.68	0.54	1	1	1
P51954	Serine/threonine-protein kinase Nek1	Nek1	136.61	32.30	0.50	1	1	1
P51410	60S ribosomal protein L9	Rpl9	21.87	31.35	3.65	1	1	1
Q8CFD5	DNA excision repair protein ERCC-8	Ercc8	43.66	31.12	1.51	1	1	1
Q9QZ85	Interferon-inducible GTPase 1	Igtp1	47.54	31.12	1.69	1	1	1
Q4ACU6	SH3 and multiple ankyrin repeat domains protein 3	Shank3	185.28	31.08	0.29	1	1	1

Response Table 1. Mass spectrometry results showing characteristics of the 32 proteins after excluding the IgG disturbance.

I am still very much in favor of publication, but homework needs to be done properly prior to that.

Re:

Thank you for your continued support of our manuscript, and we wholeheartedly agree that addressing these issues thoroughly is crucial before proceeding with publication. Your constructive feedback has provided valuable guidance, and we are dedicated to implementing the suggested improvements to enhance the quality and transparency of our study. Please do not hesitate to reach out if there are any additional aspects of the manuscript that you believe require attention.

Referee #3:

The authors have addressed some of my comments by making the proposed adjustments in their display items such as applying changes for colorblind readers as well as showing a more suitable representative image and zoom-in areas as well as providing more information in their Materials and Methods section for their quantifications.

Unfortunately however, there is no experimental data regarding my comments on the PLA assay controls nor the pyrene-labelled actin depolymerization assay. This would be of particular importance to show an improvement of the initially submitted manuscript regarding overall specificity of MICAL3 interaction as well as the following monooxygenase activity of MICAL3 on F-actin depolymerization.

Re:

Thank you for your feedback regarding the adjustments made to the display items and materials and methods section of our manuscript. We appreciate your recognition of our efforts to address your concerns and improve the clarity and accessibility of our findings.

We acknowledge your request for experimental data regarding the PLA assay controls and the pyrene-labelled actin depolymerization assay. We understand the importance of demonstrating the specificity of MICAL3 interaction and its monooxygenase activity on F-actin depolymerization. We apologize for the oversight in not including this data in the revised manuscript. We have addressed this issue now as the following according to your valuable suggestions.

1. I agree with the authors that there is good evidence shown by the Co-IP and proteomics data to claim an interaction between MICAL3 and CHK1. However, I asked for crucial internal controls for the PLA assay to show specificity of their desired antibodies and functionality of their assay, which is essential for this experimental approach and has not been shown for either the cell-based (HEK cells) PLA nor the

GV, 2-cell and 4-cell stadium. The authors refer to their experiment as strictly performed according to the manufacturers protocol. However, this protocol incorporates a Link on "how to optimize the Duolink Proximity Ligation assay" from the same company. The manufacturer even lists technical as well as biological controls that should be included in every Duolink PLA assay, similar to the ones I requested. Primary antibodies only control, as well as a positive control in HEK cells should be the minimum the authors have to do for this experiment if they want to include the PLA data in their manuscript.

2) The authors performed a PLA assay in Fig. 4 for an interaction-readout between CHK1 and MICAL3, however there are some general controls missing for the cell-based PLA. As indicated in the materials and methods the authors stained without the primary antibodies as negative control in Fig. 4D. Additionally, it would be good to see PLA assays in HEK cells with the following conditions: primary antibodies individually with either secondary antibody to exclude unspecific binding of the secondary antibody to the other primary antibody, a positive control of a known CHK1 interactor (see Blasius et al. 2011) ...

Re:

Thank you for your thorough review and valuable feedback on our manuscript regarding the interaction between MICAL3 and CHK1. We appreciate your acknowledgment of the evidence provided by the Co-IP and proteomics data supporting this interaction.

We acknowledge the importance of including crucial internal controls for the PLA assay to demonstrate the specificity of the antibodies used and the functionality of the assay especially by HEK-293 cells. We apologize for not providing these controls in the manuscript. In this revision, we have addressed this issue by including primary antibodies only controls and positive controls in HEK-293T cells, as suggested. For the positive controls, we searched the String network for known CHK1 interaction partners besides the suggested *Blasius et al. 2011* (Response Figure 3A).

We then performed the PLA assay in HEK-293T cells, including the suggested primary antibody controls and known partner controls. As expected, the blank group without any antibody and the primary antibody group with CHK1 or MICAL3 demonstrate no red PLA signal, excluding unspecific binding of the PLA signal to the primary antibodies. While the two positive control group between CHK1 and p.ATM or ATR both show relatively weak point-like signals compared with the CHK1&MICAL3 group (Response Figure 3B, we added it to Appendix Figure S2). All in all, the recommended negative or positive controls for PLA assay further verify the interaction between CHK1 and MICAL3.

We appreciate your constructive criticism and hope that all of your concerns in the revised manuscript have been addressed. If you have any further suggestions or questions, please feel free to let us know.

Response Figure 3. PLA signals imply that CHK1 can interact with MICAL3 in situ. (A) Protein-protein interaction network of CHK1 obtained from String website. Left: The protein network of CHK1; Right: The legend explains the lines in the network.

(B) Red PLA signals in HEK-293T cells indicate the interactions between CHK1 and MICAL3. The signals between CHK1 and p.ATM or ATR were used as positive controls. The z-scan model was applied to collect signals through the confocal microscope. Scale bar: 5 μ m.

2. While I do appreciate the authors explanation for quenching of the signal, I question the selection of endpoint measurements for this kinetic assay. Why was the 1 h timepoint chosen, why was the signal quenching gone by then? To me this sounds more like an optimization problem of the assay itself and this questions the actual signal the authors read out after 1 h incubation. I do not see a problem why this should not work technically as there is as an optimization guide for troubleshooting dealing with signal collection problems. It would be essential to also show actin depolymerization alone and compare the addition of wt and mutant CHK1 to actin depolymerization alone or implement a positive control for actin depolymerization as I proposed already as F-actin alone depolymerizes to a great extent after 35 min (see below Fig. 2d).

The overall experimental evidence on MICAL3 activity on actin depolymerization which is necessary for subsequent PNEB in this assay is in my opinion not sufficient. I suggest to incorporate the requested controls or to remove the panel Fig. 5 F .

Re:

Thank you for your detailed feedback regarding the endpoint measurements chosen for the kinetic assay in Figure 5F and the overall experimental evidence on MICAL3 activity on actin depolymerization. We appreciate your thorough review and your suggestions for improving the experimental design and interpretation of the results.

We acknowledge your concerns regarding the selection of the 1-hour timepoint for the kinetic assay and the necessary controls. We capture the endpoint to avoid signal quenching by too many times of fluorescence stimulus. In order to solve the optimization problem, we reevaluated the experimental conditions and tried an only F-actin group to see the depolymerization by multi-point signal collection. However,

the F-actin depolymerized at a faster speed than we expected, potentially indicating the signal quenching again (Response Figure 4).

We agree that the overall experimental evidence on MICAL3 activity on actin depolymerization may not be sufficient in its current form. We carefully considered your suggestions and decided to remove the panel Figure 5F to ensure the validity of our findings.

Thank you for bringing these important considerations to our attention. We appreciate your continued engagement and support as we strive to enhance the quality of our research.

Response Figure 4. The F-actin depolymerization result by multi-point detection in the only F-actin group demonstrates significant signal quenching.

Overall, I think the story is still very interesting and the authors have put some effort in this revision. However, the manuscript needs some controls in the above-mentioned experiments to further strengthen the authors claims on a specific CHK1-MICAL3 interaction and subsequent MICAL3-dependent F-actin depolymerization for PNEB.

Re:

Thank you for your overall positive feedback and recognition of the efforts made in the revision of the manuscript. We appreciate your acknowledgment of the potential significance of the study and your constructive critique regarding the need for additional controls. We have carefully considered your feedback and implemented the

necessary adjustments to address the gaps in the experimental design and interpretation of results.

Thank you for your time and consideration again.

Sincerely,

Shigang Zhao, M.D., Ph.D.

Center for Reproductive Medicine

Shandong University

#157 Jingliu Road, Jinan 250001, China

Tel: +86-18954109360

Email: zsg0108@sdu.edu.cn

Dear Dr. Wu,

Thank you for submitting your revised manuscript. It has now been seen by two of the original referees. My apologies for the delay in getting back to you, it took longer than anticipated to receive the referee reports.

As you can see, the referees find that the study is significantly improved during revision and recommend publication. However, I need you to address the points below before I can accept the manuscript.

- Please address the remaining minor concern of #2.
- In my previous decision letter, I had noted "During our routine figure checks, we note a potential re-use between Figure 2C F-actin and Figure EV2 F-actin and a potential cell re-use between Figure 4G DAPI and Appendix Figure S1 DAPI/merge, which is only allowed if the images/cells are derived from the same experiment, in which case it should be clarified in the legends of all respective figure panels. To which you responded "Thank you for bringing this to our attention. We carefully reviewed the images in Figure 2C, Figure EV2, Figure 4G, and Appendix Figure S1 to ensure compliance with the guidelines regarding image and cell reuse. Images in Figure 2C and Figure EV2, are indeed derived from the same experiment, we thus clarified this in the legends of Figure EV2." I appreciate that you addressed the Figure 4G DAPI and Appendix Figure S1 DAPI/merge concern. However, I am unable to locate the clarification in the legends of Figure 2C and Figure EV2. Please include statements in the legends of both Figure 2C and Figure EV2.

Thank you again for giving us to consider your manuscript for EMBO Reports, I look forward to your minor revision.

Kind regards,

Deniz Senyilmaz Tiebe

--

Deniz Senyilmaz Tiebe, PhD
Senior Scientific Editor
EMBO Reports

Referee #2:

Zhang et al. hand in their work "CHK1 controls Zygote Pronuclear EB by regulating F-actin through interacting with MICAL3" after another revision. I appreciate the use of a second MICAL inhibitor that puts this result on the safe side. I also recognize the effort to clarify the procedure to generate and interpret the CHK1 interactome by mass spectrometry, i.e. response table 1 / source data. One issue remaining: did the experiments not identify CHK1 itself and if so, why not? Finally, the figure legend may be formulated as follows:

Mass spectrometry result of the CHK1 interactome. Lysates from 6000 mouse zygotes were pooled and used for immunoprecipitation of CHK1. A total of 32 proteins were identified, possibly interacting with CHK1 in zygotes, by subtracting the proteins identified in an IgG control experiment. The unique peptide numbers and protein scores of each protein are displayed. The black dashed rectangle indicates the protein, MICAL3. For more information on the identified protein, please see the source data.

After having addressed this last issue, I am looking forward to publication of the work in EMBO Reports.

Referee #3:

The authors did a good job in revising the manuscript. They also now provide the missing controls for the PLA assays although some of the actin assays could not be performed, but this may be understandable. I recommend publication in EMBO Reports. It is a very nice study. Congratulations from this reviewer.

RE: EMBOR-2023-58263V3

Sep 3, 2024

Dr. Deniz Senyilmaz Tiebe

Senior Scientific Editor

EMBO Reports

Dear Editor and Reviewers,

We sincerely appreciate the time and effort invested by you in providing a detailed assessment of our manuscript titled "CHK1 Controls Zygote Pronuclear Envelope Breakdown by Regulating F-actin through Interacting with MICAL3 (EMBOR-2023-58263V3)", again. We have diligently addressed each of the points raised by the editor and reviewers, and we present a summary of our responses below. Comments from the editor and reviewers are in blue, while responses from us (authors) are in black. Additionally, revisions within the main manuscript are indicated in yellow. We genuinely hope that our efforts adequately meet your expectations and requirements.

Editor's and Reviewers' comments:

Editor:

In my previous decision letter, I had noted "During our routine figure checks, we note a potential re-use between Figure 2C F-actin and Figure EV2 F-actin and a potential cell re-use between Figure 4G DAPI and Appendix Figure S1 DAPI/merge, which is only allowed if the images/cells are derived from the same experiment, in which case it should be clarified in the legends of all respective figure panels. To which you responded "Thank you for bringing this to our attention. We carefully reviewed the images in Figure 2C, Figure EV2, Figure 4G, and Appendix Figure S1 to ensure compliance with the guidelines regarding image and cell reuse. Images in Figure 2C and Figure EV2, are indeed derived from the same experiment, we thus clarified this

in the legends of Figure EV2." I appreciate that you addressed the Figure 4G DAPI and Appendix Figure S1 DAPI/merge concern. However, I am unable to locate the clarification in the legends of Figure 2C and Figure EV2. Please include statements in the legends of both Figure 2C and Figure EV2.

Re:

We sincerely apologize for the oversight in not including the clarification in the legends of Figure 2C and Figure EV2. We appreciate your attention to detail, and thank you for bringing this to our attention.

In the revised version, we have included statements in the legends of both Figure 2C and Figure EV2 to clarify that the images are derived from the same experiment. This will ensure compliance with the guidelines regarding image reuse. If there are any other specific concerns or suggestions that you would like us to address, please let us know.

Referee #2:

Zhang et al. hand in their work "CHK1 controls Zygote Pronuclear EB by regulating F-actin through interacting with MICAL3" after another revision. I appreciate the use of a second MICAL inhibitor that puts this result on the safe side. I also recognize the effort to clarify the procedure to generate and interpret the CHK1 interactome by mass spectrometry, i.e. response table 1 / source data. One issue remaining: did the experiments not identify CHK1 itself and if so, why not?

Re:

Thank you for your thorough review and for raising questions regarding our interpretation of the assay results. We acknowledge that the mass spectrometry assay did not identify CHK1 itself. However, we have conducted western blot verification to identify the CHK1 protein, as demonstrated in Figure 4B and 4C. The absence of CHK1 detection by mass spectrometry could be attributed to the presence

of the IgG heavy chain, which appears at the 55 kDa position. It is important to note that the molecular weight of endogenous CHK1 is approximately 56 kDa.

We greatly appreciate your guidance and are open to any further suggestions or queries you may have. Your input is invaluable in ensuring the accuracy and robustness of our research.

Finally, the figure legend may be formulated as follows: Mass spectrometry result of the CHK1 interactome. Lysates from 6000 mouse zygotes were pooled and used for immunoprecipitation of CHK1. A total of 32 proteins were identified, possibly interacting with CHK1 in zygotes, by subtracting the proteins identified in an IgG control experiment. The unique peptide numbers and protein scores of each protein are displayed. The black dashed rectangle indicates the protein, MICAL3. For more information on the identified protein, please see the source data.

Re:

Thank you for your insightful suggestion. We have carefully considered your feedback and have updated the legend in Figure 4A of the manuscript as per your recommendations.

We appreciate your attention to detail and your valuable input. Your suggestions have greatly improved the clarity and accuracy of our figure legend. Thank you again for your valuable feedback and support.

Referee #3:

The authors did a good job in revising the manuscript. They also now provide the missing controls for the PLA assays although some of the actin assays could not be performed, but this may be understandable. I recommend publication in EMBO Reports. It is a very nice study. Congratulations from this reviewer.

Re:

Thank you for your positive feedback and recommendation for publication in EMBO Reports. We greatly appreciate your acknowledgment of our efforts in revising the manuscript and providing the missing controls for the PLA assays. We understand that some of the actin assays could not be performed, and we apologize for any inconvenience caused. We will carefully consider any further suggestions or feedback you may have to further improve the manuscript. Once again, thank you for your kind words and congratulations.

Thank you for your time and consideration again.

Sincerely,

Shigang Zhao, M.D., Ph.D.

Center for Reproductive Medicine

Shandong University

#157 Jingliu Road, Jinan 250001, China

Tel: +86-18954109360

Email: zsg0108@sdu.edu.cn

Keliang Wu
Shandong University

Dear Dr. Wu,

Thank you for submitting your revised manuscript. I have now looked at everything and all is fine. Therefore, I am very pleased to accept your manuscript for publication in EMBO Reports.

Congratulations on a nice work!

Kind regards,

Deniz Senyilmaz Tiebe

--

Deniz Senyilmaz Tiebe, PhD
Senior Scientific Editor
EMBO Reports

--
